



# Physical-biogeochemical regional ocean model uncertainties stemming from stochastic parameterizations and potential impact on data assimilation

Vassilios D. Vervatis (1), Pierre De Mey-Frémaux (2), Nadia Ayoub (2), Sarantis Sofianos (1), Charles-Emmanuel Testut (3), Marios Kailas (1), John Karagiorgos (1), and Malek Ghantous (2)

(1) University of Athens, Faculty of Physics, Athens, Greece.
(2) LEGOS/CNRS, Toulouse, France.
(3) Mercator Océan, Ramonville St. Agne, France.

*Correspondence to*: Vassilios D. Vervatis (vervatis@oc.phys.uoa.gr)

**Abstract.** We generate ocean biogeochemical model ensembles including several kinds of stochastic parameterizations. The NEMO stochastic modules are complemented by integrating a subroutine to calculate variable anisotropic spatial scales, which are of particular importance in high-resolution coastal configurations. The domain covers the Bay of Biscay at 1/36° resolution, as a case study for open-ocean and coastal shelf dynamics. At first, we identify uncertainties from assumptions subject to erroneous atmospheric forcing, ocean model improper parameterizations and ecosystem state uncertainties. The error regimes are found to be mainly driven by the wind forcing, with the rest of the perturbed tendencies locally augmenting the ensemble spread. Biogeochemical uncertainties arise from inborn ecosystem model errors and from errors in the physical state. Model errors in physics are found to have larger impact on chlorophyll spread than those of the ecosystem. In a second step, the ensembles undergo verification with respect to observations, focusing on upper-ocean properties. We investigate the statistical consistency of prior model errors and observation estimates, in view of joint uncertainty vicinities, associated with both sources of information. OSTIA-SST L4 distribution appears to be compatible with ensembles perturbing physics, since vicinities overlap, enabling data assimilation. The most consistent configuration for SLA along-track L3 data is in the Abyssal plain, where the spread is increased due to mesoscale eddy decorrelation. The largest statistical SLA biases are observed in coastal regions, sometimes to the point that vicinities become disjoint. Missing error processes in relation to SLA hint at the presence of high-frequency error sources currently unaccounted for, potentially leading to ill-posed assimilation problems. Ecosystem model-data samples with respect to Ocean Colour L4 appear to be compatible with each other only at times, with data assimilation being marginally well-posed. In a third step, we illustrate the potential influence of those uncertainties on data assimilation impact exercise, by means of multivariate representers and EnKF-type incremental analysis for a few members. Corrections on physical properties are associated with large-scale biases between model and data, with diverse characteristics in the open-ocean and the shelves. The increments are often characteristic of the underlying mesoscale features, chlorophyll included due to the vertical velocity field. Small scale local corrections are visible over the shelves. Chlorophyll information seems to have a very measurable potential impact on physical variables.

## 1    Introduction

The need for specific Data Assimilation (DA) methods and sustained observations in regional models and in coastal regions of larger-scale models has been identified early enough (see extensive reviews in *De Mey,* 2000 and *Pinardi et al.,* 2017). Continental shelf and slope seas differ from the open ocean in the presence of the coast, strong bathymetry gradients, inputs from rivers, and shallower water. Flows in these shallower seas are forced by (inter alia) pressure and current fields from ocean-scale mass balances, circulation, tides and eddies, winds and air pressure variations, non-uniform density. For all





these, responses differ between the deep ocean and shallower shelf, and *"the influence of coastal ocean processes is felt far beyond shelf break, interacting with open ocean dynamics and controlling the connectivity of remote ecosystems"*[1].

In the coastal parts of regional ocean models, many factors can complicate the assimilation of data compared to the deep-ocean; to name but a few, free-surface variations (tides, storm surges), anisotropy (offshore scales are generally shorter than

alongshore scales), non-homogeneity, friction and mixing effects throughout the water column (driven in part by tides), and the fast response to the atmosphere on the shelf. The characterization and specification of model error are critical in any assimilation scheme, but extremely challenging in the coastal zone. The model errors are strongly dependent on time scale, but any attempt at separation is confounded by strong nonlinearity in the dynamics that can couple variability at different frequencies.

As a first step, one must characterize the forecast errors under various error regimes by methods which include realistic error dynamics such as stochastic modelling (for example some recent studies for regional configurations are *Vandenbulcke and Barth*, 2015, in review 2018; *Quattrocchi et al.,* 2014; *Vervatis et al.,* 2016 and references within). Most studies point at the benefit of "advanced" assimilation methods with built-in error propagation (*Kourafalou et al.,* 2015a, b), such as the Ensemble Kalman Filter (EnKF; *Evensen*, 2003) and other ensemble methods. As pointed out in *Schiller et al.* (2015), there

may not be enough observations in many regions of the world to reliably estimate uncertainties of numerical models, to assimilate in those models, and to enable deterministic forecasts. Therefore, it can be expected that Ensemble Prediction Systems (EPS) will be widely used in the future, complementing the "deterministic" approach, for quantifying uncertainties in coastal products, and for providing probabilistic forecasts.

Marine biogeochemical DA is progressively used in operational platforms as a tool to improve ocean forecasting systems;

however, this remains a maturing subject with several challenges still to overcome. The EnKF in a simple 1D ecosystem model was first introduced by *Eknes and Evensen* (2002) and later by *Allen et al.* (2003), controlling the evolution of zooplankton and nutrients by assimilating chlorophyll. *Simon and Bertino* (2009) extended the EnKF to include a Gaussian anamorphosis transformation, accounting for non-Gaussian biogeochemical distributions. *Ciavatta et al.* (2014) instead of assimilating chlorophyll, adapted the EnKF to assimilate the SeaWiFS light attenuation coefficient $K_d(443)$, incorporating a

bio-optical model. Moreover, recent studies have shed light on the assimilation of Ocean Colour Plankton Functional Types (PFTs), further improving marine ecosystem simulations (*Ciavatta et al.,* 2016, 2018). Overall, biogeochemical model performance is strongly dependent by ocean dynamics and by several options in the assimilation scheme, and so the use of such systems is of vital importance for advancing ecosystem models (*Gehlen et al.,* 2015).

*Lucas et al.* (2008) performed stochastic ensembles using NEMO (Nucleus for European Modelling of the Ocean;

http://www.nemo-ocean.eu/; *Madec*, 2012), based on a NATL4 configuration for the North Atlantic at 1/4° resolution. In their study, perturbations were derived based on EOF modes, in a multivariate context for air temperature and wind velocities, and in univariate for the incoming solar radiation. Later on, tools and methodologies to generate ensembles of perturbed NEMO simulations were introduced, including several kinds of stochastic parameterizations to simulate unresolved processes and scales. The ensemble capabilities of NEMO have been discussed in the literature following the

SANGOMA (http://www.data-assimilation.net/) and OCCIPUT projects (*Penduff et al.,* 2014), focusing on global and regional academic configurations spanning from 2° to 1/4° resolution (i.e. ORCA2, eORCA025, eNATL025) and from seasonal to decadal time scales (*Brankart,* 2013; *Brankart et al.,* 2015; *Candille et al.,* 2015; *Garnier et al.,* 2016; *Bessières et al.,* 2017).

The scientific objectives of this paper cover a broad spectrum of interdisciplinary components, focusing on the generation of

ensembles in high-resolution regional models. The study aims at guiding future ensemble-based modelling strategies, in

---

[1] In : *De Mey and Kourafalou* (2014)





support of ensemble-based DA and probabilistic forecasting approaches. There are two scientific questions addressed in this work, using the Bay of Biscay as a case study for open-ocean and coastal processes. On the basis of prior knowledge in the literature for stochastic approaches and on what is feasible in terms of computational resources, we have selected a few sources of model uncertainties and we have excluded others, investigating: (a) Which are the main physical-biogeochemical

model uncertainties in regional/coastal systems and how can we estimate them? (b) What are the impacts of the choice of different set of physical-biogeochemical perturbations on the analysed ocean state, by means of multivariate ensemble-based DA? In order to answer the first question, in Section 2, we present a generic method to complement the NEMO stochastic modules for high-resolution configurations, following the work of *Brankart* (2013), *Brankart et al*. (2015) and *Bessières et al*. (2017). In Section 3, we explore possible sources of ocean-ecosystem model errors in the coupled system and present an

ensemble-based innovation statistics framework to evaluate upper-ocean uncertainties. The second question is addressed in Section 4, illustrating the potential impact of those ensembles on data assimilation by means of multivariate representers and EnKF-type incremental analysis for a few members. The concluding remarks of this work, including the code and data availability, are summarized in Sections 5 and 6.

## 2         Methodology and experimental design

**2.1        The coupled ocean–biogeochemical model**

In this study, we use a regional configuration based on NEMO, in its stable version 3.6, covering the Bay of Biscay and the western part of the English Channel (Fig. 1). For a complete description of the numerical set-up, identical to the IBI-MFC within CMEMS, and for details of validation, the reader is referred to *Maraldi et al*. (2013).

The NEMO ocean engine OPA (Océan PArallélisé) is coupled on-line with the passive tracer package TOP2 and the

biogeochemical model PISCES-v2 (Pelagic Interactions Scheme for Carbon and Ecosystem Studies volume 2; *Aumont et al*., 2015). The high-resolution configuration (hereafter BISCAY36) is inherited from earlier studies during MyOcean (*Quattrocchi et al*., 2014; *Vervatis et al*., 2016). The meteorological fields are provided by ECMWF (European Center for Medium-Range Weather Forecasts). The initial state and the components of the ocean and biogeochemical open boundary conditions, are inquired through the daily and weekly archives of CMEMS infrastructure, respectively. For the ocean model

set up the reader is referred to *Vervatis et al*. (2016).

The open boundaries of the biogeochemical model are forced by the global system BIOMER4V1R1 (resolution: 1/2°; http://www.mercator-ocean.fr/), providing 3D global weekly mean analysis of dissolved iron, nitrate, phosphate, silicate, oxygen, chlorophyll, phytoplankton concentrations and primary production parameters. The coupling frequency between the ocean model and PISCES is set one every two time-steps, i.e. 150 s for physics and 300 s for biogeochemistry. The on-line

high-frequency coupling is optimal compared to an off-line approach, in terms of conservation of tracers. Note that on-line coupling means one-way ocean forcing to the ecosystem model, since no feedback is given back to the circulation model. The primitive equations are discretized on a 1/36° curvilinear Arakawa C-grid and the tracer transport model runs also in the same grid resolution (no coarsening). The TOP2 package controls the advection-diffusion equations of the passive tracers and the biogeochemical SMS (Sources Minus Sinks) terms. The numerical scheme for the biogeochemical processes is

forward in time (Euler) and differs from the classical leap-frog scheme used in physics. The advection scheme is the same as in physics, i.e. QUICKEST (*Leonard*, 1979), but using limiter of *Zalezak* (1979). These options have been tested by *Gutknecht et al*. (2016) and now are used in the IBI-MFC operational system (http://marine.copernicus.eu/).

The equations of PISCES include 24 prognostic variables simulating the biogeochemical cycles of oxygen, carbon and the main nutrients controlling phytoplankton growth, i.e. nitrate, ammonium, phosphate, silicic acid and iron. The model

distinguishes four plankton functional types based on size, including two phytoplankton compartments (nanophytoplankton





and diatoms) and two zooplankton classes (microzooplankton and mesozooplankton). The distinction of the two phytoplankton size classes, along with the description of multiple nutrient co-limitations allows the model to represent ocean productivity, across different biogeographic ocean provinces (*Longhurst*, 1998). The phytoplankton prognostic variables are the total biomass in C, Fe, Si (only for diatoms) and chlorophyll and hence the Fe/C, Si/C, Chl./C ratios are variable. This

allows a more accurate conversion of phytoplankton into chlorophyll concentrations, which is of great importance for comparisons with proxy ocean colour satellite data. Other Redfield ratios between C/N/P are kept constant according to *Takahashi et al.* (1985) values 122/16/1. PISCES also distinguishes three non-living pools for organic carbon: small Particulate Organic Matters (sPOM), big Particulate Organic Matters (bPOM; different settling velocities with sPOM) and semi-labile Dissolved Organic Carbon (DOC).

## 2.2 Stochastic parameterizations

A generic approach based on first-order autoregressive processes AR(1) of different stochastic parameterizations is proposed for both components of the coupled system. The work is based on recent advances in NEMO explicitly simulating the effects of model uncertainties. A comprehensive analysis for the stochastic formulation of NEMO is given in *Brankart* (2013) and *Brankart et al.* (2015). A theoretical background for probabilistic ocean modelling, with technical details on implementation

strategies based on NEMO (e.g. online ensemble diagnostics, connection with observation operators and DA systems) is provided by *Bessières et al.* (2017).

*Brankart et al.* (2015) studied the impact of two broad categories in ensemble modelling: the first is the Stochastic Perturbed Parameterized Tendencies (SPPT) and the second is the Stochastic Parameterization of Unresolved Fluctuations (SPUF). The SPPT implementation aims at estimating relevant perturbations on the models' parameterized tendencies and perform

Monte-Carlo techniques to obtain a pdf of these tendencies. The stochastically derived parameterized tendencies are added to the models' non-parameterized tendencies (the latter assumed free of uncertainties). In line with SPPT, the SPUF implementation aims at obtaining an ensemble of forecasts with enhanced reliability. The method is based on random walks sampling gradients from the state vector. Those gradients are added to the models' operator as unresolved scales and/or (bio)diversity.

In this study, ensemble simulations are performed with the SPPT method in which stochastic perturbations are generated by maps $\xi$ of AR(1) processes. In practice, every time-step Gaussian AR(1) processes are generated following the expression defining $\xi$:

$$\xi_{k+1} = e^{-\frac{1}{\tau}} \cdot \xi_k + \left( \sigma \cdot \sqrt{1 - e^{-\frac{2}{\tau}}} \right) \cdot w + \mu \cdot \left( 1 - e^{-\frac{1}{\tau}} \right) \qquad (1)$$

where $k$ is the model time-step, $w$ is white Gaussian noise, $\mu, \sigma$ and $\tau$ the mean, the standard deviation (uncertainty

amplitude) and the correlation timescale, respectively. *Brankart et al.*, (2015) performed independent Gaussian autoregressive processes at every model grid point and introduced spatial dependence via Laplacian filtering to the $\xi^{(i)}$ 2D or 3D maps (grid index $i = 1, ..., m$). In their work, an alternative is also proposed to solve an elliptic equation noting that both methods are flow dependent.

In the present study, we aim at complementing the NEMO stochastic modules, by integrating a subroutine to calculate

explicitly variable anisotropic 2D spatial scales. The latter is of high importance, especially in high-resolution regional applications. Technically, this is done by solving an elliptic Gaussian equation on a few model grid points randomly selected in the domain. In general, the multivariate normal distribution expression $\mathcal{G}(r)$ is given by:

$$\mathcal{G}(r) = \frac{1}{\sqrt{|2\pi\Sigma|}} e^{-\frac{1}{2}(r-\bar{r})^T \Sigma^{-1}(r-\bar{r})} \qquad (2)$$





where $\boldsymbol{r} \in \{r_1, \dots, r_s\}$ a distance vector from the mean centers $\bar{\boldsymbol{r}} \in \{\overline{r_1}, \dots, \overline{r_m}\}$ located on a few model grid points $i = 1, \dots, m < s$, and $\boldsymbol{\Sigma}$ the covariance matrix $\mathbb{R}^{m \times m}$ (in general non-diagonal for rotated distributions) of the variances $\boldsymbol{\sigma_r}^2 \in \{\sigma_{r_{11}}^2, \dots, \sigma_{r_{mm}}^2\}$ controlling the length scales of the Gaussian distributions. The vector $\boldsymbol{r}$ covers the whole model domain over which perturbations are generated. For this specific configuration, we use a bivariate form of the elliptic Gaussian equation (i.e. 2D spatial fields), with diagonal $\boldsymbol{\Sigma}$ covariance matrix. Other options include a tensor form that would make those patterns coast-aware (*Barth et al.*, 2009). The latter is not important for the specific BISCAY36 configuration (e.g. not many islands/islets) and for the variables we choose to perturb (e.g. atmospheric forcing). The relation between $\xi$ and $\mathcal{G}(\boldsymbol{r})$ maps is illustrated in the programming flowchart for the integrated algorithm in Appendix A.

The tuning of the perturbation patterns depends on the properties of each parameterized tendency. A typical correlation length can be chosen per tendency. In order to introduce anisotropy, the spatial length scales and the mean vector of the 2D distributions vary randomly per ensemble member. Instead of implementing white noise $w$ in Eq. (1), a multi-modal stochastic pattern is calculated as the sum of Eq. (2) over a few randomly selected model grid points solving the equation:

$$\sum_{i=1}^{m} \mathcal{G}(\boldsymbol{r})^{(i)} = 0 \tag{3}$$

The uncertainty amplitude of the AR(1) processes can be tuned by tapering techniques at $\pm \sigma$ declared in Eq. (1). The method incorporates the anamorphosis transformation capabilities of NEMO discussed in *Brankart et al.* (2015). This is done, by applying a nonlinear change of the Gaussian stochastic maps into non-Gaussian distributions, before using them in the models' parameterized tendencies.

Overall, this work provides a simple and computationally inexpensive way introducing spatial scales in comparison to the filtering operator. The implementation is done in a full MPI context and the technical aspects are presented in Appendix A. The method is generic and yields positive results, in terms of realistic spatial patterns compared to noise, especially in high-resolution configurations. Figure 2, shows examples of AR(1) stochastic patterns drawn for several variables of the coupled system, in order to introduce model uncertainties. In the case of the elliptic Gaussian equation a robust stochastic pattern is explicitly formulated, such as for instance the wind, with multi-scales varying around an average of $\sigma_r \sim 1°$ (Figs. 2a-d). In the same simulation larger spatial stochastic patterns can be applied to other atmospheric variables, for example the air temperature and sea level pressure adopting scales of $\sigma_r \sim 2°$ and $\sigma_r \sim 3°$ respectively (Figs. 2e-f). The optimal use of the filtering operator is achieved in coarse global/regional configurations performing a few Laplacian passes (*Brankart et al.*, 2015; *Garnier et al.*, 2016). At 1/36° resolution the option to iterate 100 times the Laplacian operator results to noisy spatial patterns, not representative for most oceanic processes (Fig. 2g).

### 2.3    Experimental environment

Ensemble forecasting systems are optimally designed in view of high demand in computational resources. The present configuration uses the enhanced MPI strategy of NEMO for double parallelization in the spatial domain and the ensemble dimensions (*Bessières et al.*, 2017). The latter means that ensemble simulations are carried out by just one call to the executable of the coupled NEMO-PISCES system. The approach is generic and provides the flexibility to test various options in the resources geometry, under different programming environments and HPC facilities. In this study, BISCAY36 scales-out using 96 processors of domain decomposition per ensemble member, excluding land processors. The configuration uses the NEMO I/O strategy and is connected to an external server (i.e. XIOS controlled by an XML file) thus, increasing the total number of processors for the ensemble simulations including those handling the I/O specifications (e.g. model variables, domains, grid, output frequencies etc.). In this context, our ensemble experiments are designed to fit the scalability limits encountered in nowadays operational systems. *Vervatis et al.* (2016) showed that results for BISCAY36 converge with increasing ensemble size in the range of 20 to 40 members. In this work, we set the same range of members





resulting to a scalability problem of the order of $O(10^3)$ cores (cf. Appendix B). The operational feasibility of such a system is viable in today's NWP systems and elements of this work could be transferred to other operational platforms.

A five months' spin-up period is allowed for the free run with no stochastic parameterizations (hereafter Control Run-CR), from July to November 2011. As in *Vervatis et al.* (2016) the CR is carried out for the ocean model to develop coherent

structures starting from PSY2V4R2 analysis (resolution: 1/12°; http://www.mercator-ocean.fr/) and identify the main physical processes in the Bay of Biscay. The same period is used for the spin-up of the on-line coupled biogeochemical model. The deterministic CR is extended from December 2011 to June 2012 and serves as a reference for the ensemble experiments. In Fig. 3, we present the ocean state of the unperturbed CR on April 30, 2012, for the surface variables SSH, SST, SSS, total chlorophyll concentration and the two classes of chlorophyll "nano" and "diatoms".

One important challenge in ensemble forecasting is to find the most important sources of model errors and assess properly the stochastic parameterizations. In order to tackle this issue, we designed a twofold experiment. At first we aim at identifying model uncertainties in the Bay of Biscay, performing medium-range ensembles of 20 members for one month during the spring bloom period in April 2012 (i.e. S1-8 experiments in Table 1). Several stochastic parameterizations are tested on a case-by-case basis. The stochastic options are reassessed for each sensitivity experiment. A final selection of

perturbations is established, having a noticeable impact in terms of spread in upper-ocean properties. In a second step, we perform seasonal-range ensembles of 40 members (i.e. Ens1-3 experiments in Table 1) following an optimal stochastic environment, defined from the previous less expensive computationally sensitivity experiments. In line with this, we decided not to perturb variables that the model is not sensitive to, as for example the Photosynthetically Active Radiation (PAR) coefficient $k_{PAR}$ for the penetrative solar radiation (sensitivity experiments not shown). In summary, the stochastic

parameterizations are optimal in terms of realistic physical-biogeochemical perturbations, i.e. proper spatiotemporal scales and noise-to-signal ratio, as well as computational efficiency (e.g. no members are dropped during the ensemble run). The seasonal-range ensembles are carried out from December 2011 to June 2012 and focus on the impact of model errors in surface chlorophyll concentrations.

In Table 1, we summarize our adopted options for an optimal stochastic protocol of the sensitivity experiments S1-8 and

seasonal-range ensembles Ens1-3 in BISCAY36, including also Ens0 as a reference ensemble discussed by *Vervatis et al.* (2016). The Ens0 ensemble is different from the Ens1-3 ensembles, since it was generated by performing stochastic modelling of the wind forcing using EOF modes. All sensitivity S1-8 and seasonal-range Ens1-3 ensembles (and their members) were initialized by using the ocean and the biogeochemical states of the deterministic CR, without perturbing the initial conditions. Flow dependent errors are not constrained by DA and the perturbation mechanism remains at work

through the whole period. A total number of 40 stochastic restarts is archived, where "pseudo-random" seed numbers (with different random sequences) are saved for all stochastic parameterizations. The production of those 40 stochastic restarts is achieved by performing a pre-simulation experiment of only a few time-steps, using the NEMO ensemble capabilities in double parallelisation for the model domain and the ensemble members. Those stochastic restarts were used to ensure the reproducibility of the stochastic patterns upon initialisation of the ensembles. The latter capability allows us to investigate

the growth rate of model uncertainties, with the same age of errors, under different atmosphere-ocean states. For this, we use the first 20 stochastic restarts to initialize the sensitivity experiments S1-8 in April 2012, as well as to initialize the first 20 members of the seasonal-range ensembles Ens1-3 in December 2011. The latter exercise aims at generating the same stochastic patterns for 20 members, over two different periods. In addition, the second set of 20 members is used to test the convergence of covariances in our seasonal-range ensembles.

In this study, we investigate whether ensemble-based cross-covariances are enriched, with the possibility to increase DA performance, by perturbing both components of the coupled ocean-biogeochemical model. In this regard, an increasing





complexity of experiments to augment chlorophyll spread is designed as follows: initially perturbing only physics (Ens1), then only biogeochemistry (Ens2) and finally both models simultaneously (Ens3). Following our strategy with the stochastic restarts, we are able to reproduce the ocean circulation of the Ens1 40 members in both Ens2 and Ens3 experiments. The same applies for the Ens2 biogeochemical perturbations integrated in Ens3 run. In the coupled simulation, the evolution of

the biogeochemical tracers is described by the advection-diffusion equation:

$$\partial_t C = \overbrace{\underbrace{- \nabla (u \cdot C)}_{advection} \overbrace{- K_h \nabla_h^2 C}^{Ens1} \underbrace{- \partial_z (K_z\, \partial_z C)}_{\substack{vertical \\ diffusion}}}^{Ens3} + \overbrace{\underbrace{SMS(C)}_{biology}}^{Ens2} \tag{4}$$

where on the right hand of Eq. (4) the first term represents the advective transport of tracers along isopycnals, the second and third terms the 3D parameterized diffusion processes and the last term denotes all biological processes affecting the concentration of tracer $C$ including the Sources Minus Sinks ($SMS$), such as for example respiration, death and grazing in

phytoplankton growth and decay. The experimental protocol followed herein is illustrated in Eq. (4) with the production of different ensembles.

### 2.4      Generation of stochastic perturbations

Model parameterized tendencies $\partial_t \mathcal{P}(\boldsymbol{x}, \boldsymbol{u}, \boldsymbol{p})$ are a function of the state vector $\boldsymbol{x}$, the forcing $\boldsymbol{u}$ and the vector of the model parameters $\boldsymbol{p}$. The proposed experimental protocol is designed to reveal the main sources of model uncertainties. In addition,

we aim at expanding the ensemble capabilities of BISCAY36 compared to *Vervatis et al*. (2016) wind perturbations. The SPPT-AR(1) approach is now applied to several kind of model tendencies. These uncertainties emerge from assumptions on erroneous atmospheric forcing, ocean model improper parameterizations and ecosystem model state uncertainties. The scientific base of these assumptions is discussed below.

### 2.4.1      Erroneous atmospheric forcing

Downscaling methods are subject to many sources of model uncertainties and one of the most prominent is the boundary conditions. In this study, we consider uncertainties based on atmospheric forcing, though equally important are errors imposed on open-ocean boundaries (not discussed in this study; the subject is investigated by *Ghantous et al*., in review 2018). The atmospheric forcing in coastal/regional applications is likely to trigger large scale biases and constitute a major source of ocean model uncertainties. We investigate here uncertainties on the wind velocities $\boldsymbol{U} = (\boldsymbol{u}_{air}, \boldsymbol{v}_{air})$, the Sea

Level Pressure ($\boldsymbol{SLP}$) and the air temperature ($T_{air}$), i.e. $\boldsymbol{u} \in \{\boldsymbol{U}, \boldsymbol{SLP}, T_{air}\}$. The ECMWF fields are multiplied by AR(1) stochastic processes (Figs. 2a-f) following an SPPT scheme:

$$\boldsymbol{u}_k \rightarrow (1 + \alpha \cdot \xi_k) \cdot \boldsymbol{u}_k \tag{5}$$

at every time-step $k$, where $\alpha$ is an optional tapering value in the range [0 1] (*Buizza et al*., 1999). In Eq. (1), we select representative values for the uncertainty amplitude $\sigma$, the average spatial correlation length $\sigma_r$ and the correlation timescale $\tau$

per tendency. Output diagnostics from the CR are analysed to deduce pdfs (not shown) for all variables of interest. Wind $\boldsymbol{U}$ and $\boldsymbol{SLP}$ exhibit normal distributions, whereas $T_{air}$ shows a bimodal distribution due to the seasonal cycle. Hereof, we assume that uncertainties are related to synoptic timescales, e.g. atmospheric phenomena such as storms, and we set a temporal correlation length of a few days for all atmospheric variables. The synoptic timescales are also verified by time-lagged autocorrelation methods applied in the CR. The spatial scales of the atmospheric fluctuations are determined by

calculating the atmospheric perturbations over synoptic timescales or by performing EOFs (*Vervatis et al*., 2016 and references within). The signal-to-noise ratio is assigned according to the statistical properties of the pdfs. The values reported in Table 1 are in agreement with other studies in the literature (e.g. *Palmer et al*., 2009).





### 2.4.2 Ocean model improper parameterizations

Air-sea fluxes of momentum, heat and mass are key quantities linking the two mediums. The physical processes related to these quantities are parameterized in terms of bulk coefficients. These parameterizations are deduced from empirical laws and incorporate wind-speed dependent coefficients, with feedback from the sea state on the fluxes. In this study, we assume

model errors based on limitations of the parameterization of air-sea interaction. Stochastic perturbations are imposed to the models' momentum drag $c_d$, latent $c_e$ and sensible $c_h$ heat coefficients. The AR(1) distribution and temporal scales are the same as those of the wind (Table 1). On the other hand, the spatial scales are assumed to be the one of the ocean state and are set of a few Rossby radii of deformation in BISCAY36 (Fig. 2g). The same stochastic pattern is applied to all coefficients, i.e. the coefficient perturbations are dependent to each other, in order to retain the convergence of iterative processes in the

CORE bulk formulae (*Large and Yeager*, 2004) and to augment the impact of the method. The positiveness of the coefficients (similarly to lognormal distributions) is verified by tapering methods and for different stability conditions and wind speed regimes in the bulk formulae. Similarly to Eq. (5) the SPPT perturbation scheme is expressed as:

$$\boldsymbol{p}_k \rightarrow (1 + \alpha \cdot \xi_k) \cdot \boldsymbol{p}_k \tag{6}$$

Model errors based on flux boundary conditions are also imposed at the bottom layer. The bottom drag $c_b$ parameterizations

are based on model assumptions for the vertical shear, the mixing scheme and the nature of the seabed (rocky, sandy or muddy), which modifies the bottom boundary layer. The stochastic fluctuations aim at introducing model uncertainties due to tidal mixing in the shelves. The bottom drag in many cases is approximated as a permanent feature in ocean models (e.g. constant minimum values in the Abyssal plain as in *Maraldi et al*., 2013) and therefore, large temporal scales up to one month are imposed in Eq. (1). Finally, considering limited knowledge for the dominant scales in the bottom layer (or more

precisely, the departures from the "mean bathymetry" involving the nature of the seabed), we apply white noise and Laplacian filtering to introduce AR(1) spatial scales (Fig. 2h). The formulation of the bottom drag follows a quadratic log-law, with minimum positive values clamped at $2.5 \times 10^{-3}$ in the Abyssal plain and maximum values observed in the shallow areas of the English Channel.

### 2.4.3 Ecosystem model state uncertainties

Marine ecosystems encompass many sources of uncertainties stemming from unresolved model processes. These uncertainties are attributable to two broad categories, i.e. unresolved biodiversity and unresolved scales. The first category refers to the biodiversity restriction of only a few tens of species resolved by the model in an effort to reduce state variables (*Le Quéré et al*., 2005). This case calls also attention to errors in the parameterization of biogeochemical missing processes, controlling the feedback between sub-systems, i.e. ecosystem, chemistry, oxygen and carbonate models. These errors emerge

from a limited number of compartments, often leading to a crude parameterization of their processes. The second category indicates uncertainties of unresolved scales, imposed even by resolved species. The SPUF scheme appears to be the most natural method to simulate uncertainties for both categories of unresolved biodiversity and scales. However, one must consider that performing random walks in the view of a large state vector could be computationally expensive. In line with this, unresolved biodiversity can be explored via the SPPT scheme (*Brankart et al*., 2015; *Garnier et al*., 2016).

Model errors due to unresolved biodiversity can be investigated by perturbing the biogeochemical tracers and/or the parameters (e.g. nutrient limitations, growth/mortality rates, grazing etc.). In this study, we follow the work by *Brankart et al*., (2015) and use an SPPT-AR(1) scheme to introduce uncertainties in all 24 tracers of PISCES, in the $SMS(C)$ term of Eq. (4). This is practically done by introducing a stochastic field $\xi_k$ described in Eq. (1), at every time-step $k$, given by:

$$SMS_k(C) \rightarrow SMS_k(C) \cdot e^{\left(\xi_k - \frac{\sigma^2}{2}\right)} \tag{7}$$



The stochastic perturbations are assumed to have a lognormal distribution and large uncertainty amplitude $\sigma$ up to 60% (Table 1). The bias correction term $\frac{\sigma^2}{2}$ (*Simon and Bertino*, 2009) is part of the model tuning to increase patchiness in the areas near peak values (Fig. 2i). The spatial scales are representative of a few Rossby radii of deformation and correlation time-scale up to 10 days, i.e. characteristic of the underlying mesoscale dynamics. Note, that these are 3D stochastic

fluctuations. Sensitivity experiments showed that perturbing all tracers across all levels with the same 2D stochastic pattern, yields robust uncertainty regimes with an ensemble spread increasing in time. The latter approach is the one followed in this study. Alternatives such as having different stochastic patterns per level and/or tracers degrades the impact of the method.

### 3 Ensemble generation and evaluation framework

#### 3.1 Like-for-like ensemble simulations

Ensemble like-for-like sensitivity experiments S1-8 are performed in order to assess error regimes based on stochastic modelling of the ocean-ecosystem state. In Fig. 4, we present maps of model uncertainties for all medium-range ensembles after one-month spin-up on April 30, 2012. The reference simulation for those second-order moments is the CR, illustrated in Fig. 3, for the upper-ocean state variables under investigation. These small ensembles are useful for the tuning of the stochastic parameters and their spread is moderate compared to the seasonal-range ensembles with longer spin-up period.

Results in medium-range experiments suggest that error regimes for SST and SSH are mainly driven by the wind forcing (Fig. 4a, g in comparison with Fig. 4f, l). Wind uncertainties have a large impact on upper-ocean uncertainties in terms of Sverdrup dynamics influencing both geostrophic and Ekman components. Imposing the same perturbation field for both u and v wind velocities, i.e. not changing the wind azimuth, result in similar uncertainties for the vorticity and Ekman pumping, further enhancing model errors.

The rest of the perturbed variables locally augment the ensemble spread in filament-like patterns in the periphery of eddies, near river plumes and in the shelf slope due to energy trapping (cf. Fig. 4 with respect to Fig. 3). Air temperature uncertainties have a significant impact on SST (Fig. 4b) and chlorophyll compared to other experiments excluding S1 (Fig. 5). Uncertainties in the wind drag coefficient have moderate impact on the wind stress and consequently on the spread of surface variables compared with those of the wind forcing (approx. an order of magnitude smaller; Fig. 4d, j). This is

because the expression of $c_d$ has a correction dependency based on the different wind speed regimes. The ocean response to SLP forcing has two components: the first-order Inverse Barometer (IB), which is isostatic and dominant at large scales, and the second-order non-IB, which depends mostly on the geographic region. The IB response to SLP perturbations has spatial scales that are equal or larger than the external Rossby radius and the IB pumping on the Abyssal plain (Tai, 1993). If we subtract the IB response from SSH then we are left with the non-IB, which in our case is of limited impact on SST and SSH

(Fig. 4c, i). Uncertainties in the bottom drag coefficient amplify error regimes mostly for SST and less for SSH, along the shelf break and on the shelves, and especially on the macrotidal area of the English Channel dynamically controlled by strong tidal currents and fronts (Fig. 4e, k).

Biogeochemical uncertainties arise from inborn ecosystem model errors and from errors in the ocean state variables (Fig. 5). All sensitivity experiments perturbing physics leave an imprint on chlorophyll uncertainties, which is on several occasions

significant compared to other model variables, like for instance the SSH and SST. When perturbing only the ecosystem model, we implement an identical stochastic pattern across all variables and vertical levels in order to increase the impact of the method. Since physics are not perturbed, the biogeochemical uncertainties are passively advected via ocean circulation. In all cases, physical model errors are found to have larger impact on chlorophyll spread than those of the ecosystem perturbations. The latter is explained by the fact that ocean physics identical to all members (i.e. S7) is a strong dynamical

attractor to the system, by means of a nutrient export heavily influenced by the velocity field, which consequently tends to

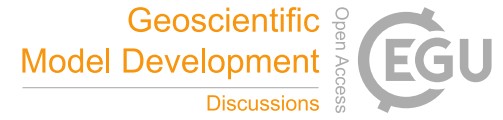



supress other sources of uncertainty, such as those of the SMS of the biogeochemical model. In general, the spread is largest in S8 where both physics and biogeochemical perturbations are applied simultaneously. Biogeochemical uncertainties result from errors in the coupled system in variable proportions depending also on the location. An example, is the chlorophyll uncertainties in the English Channel compared to the mesoscale field in the Abyssal plain.

An uneven ensemble spread is apparent comparing "nano" and "diatoms" compartments of chlorophyll (Figs. 5i-n). The latter is expected from the different model parameters of the phytoplankton classes and the additional requirements in nutrient supply of Si for the primary production of diatoms. Larger uncertainties between the two classes are observed for the nano-chlorophyll, especially in the open-ocean, whilst diatoms have larger uncertainties in the shelves and in the English Channel, where errors are dynamically controlled by tides. In line with this, it is evidenced that the uncertainty regimes for

each class follow the different size class chlorophyll abundance patterns in the region (cf. Fig. 5 with respect to Fig. 3), especially during spring bloom.

An interesting remark is that even though model uncertainties are in general larger when both physical and biogeochemical models are perturbed, a small decrease in chlorophyll spread may be observed over the Abyssal plain in presence of coherent eddies. This effect is attributed to two facts. First, the ensembles are not constrained by DA and therefore, "old-errors" in the

system contaminate the error regimes of "flow-dependent" errors and second, biogeochemical processes may be resolved on the sub-mesoscale cancelling part of the mesoscale error patterns. The latter does not change the fact that the dominant spatial scales in the Abyssal plain are characteristic of the underlying quasigeostrophic mesoscale features, chlorophyll included due to the mesoscale vertical velocity field.

The investigation of uncertainty regimes for other state variables of PISCES, is beyond the scope of this study. A brief

remark, is that the observed error patterns in upper-ocean ecosystem properties, for instance for zooplankton classes or nutrients, are similar to those of chlorophyll classes (not shown).

### 3.2    Ensemble-based innovation statistics

The ensembles undergo verification with respect to observations and in particular with respect to model-data misfits (hereafter called "innovation" for simplicity). In this study, we focus on upper-ocean properties for SLA, SST and surface

total chlorophyll. First, it is understood that the innovation spread is the result of prior uncertainties of both the model and observations. Therefore, that spread should be consistent in terms of ensemble statistics with the prior model uncertainty estimates and the observational uncertainties (measurement and representativity errors). The observational networks are accessed via the CMEMS TACs infrastructure (http://marine.copernicus.eu/) and are summarized in Table 2. The observational errors are drawn from literature or the TACs infrastructure when available, and their estimate is set to a

representative constant value for each variable (Table 2).

For the consistency analysis of Ens1 perturbing physics only, we use two observational networks, one for SST and one for SLA (Obs1a, b in Table 2). In turn, one observational network is used for the consistency check of Ens2 and Ens3 ecosystem model uncertainties (Obs2 in Table 2). The objective is to compare distributions, calculating first- and second-order statistics in data space pertaining to the distribution of observational samples, ensemble samples in data space and innovation samples,

i.e. "Observations minus Ensemble" ($OmE$) metrics. For this task, observations are perturbed with a Gaussian random number, in order to generate data distributions with the proper error standard deviation for each network. All observations are considered independent (no cross-correlations) and their Error Covariance Matrix (ECM) is diagonal.

Considering an observation $\boldsymbol{y}^o$ and model states $\boldsymbol{x}^f$ of the ensemble in data space, i.e. $H(\boldsymbol{x}^f)$, we define the innovation vector $\boldsymbol{d} = \boldsymbol{y}^o - \boldsymbol{y}^f = \boldsymbol{y}^o - H(\boldsymbol{x}^f) \approx \varepsilon^o - \boldsymbol{H}\varepsilon^f$, where $H$ an observation operator and H a linearized version, $\varepsilon^o \in N(0, \boldsymbol{R})$

the measurement error, with $\boldsymbol{R}$ being the diagonal observational ECM, and $\varepsilon^f \in N(0, \boldsymbol{P}^f)$ the true (unknown) model error,





with $\boldsymbol{P^f}$ being the prior ECM. The approximated form $\widehat{\boldsymbol{P^f}}$ of the prior ECM is derived by the estimate of the model error $\widehat{\varepsilon^f} = \boldsymbol{x^f} - \overline{\boldsymbol{x^f}}$. The following ensemble-based consistency statistics in data space are calculated:

Ensemble spread (i.e. 1std):  $\qquad \sigma_f = \sqrt{\overline{(\boldsymbol{H(x^f)} - \overline{\boldsymbol{H(x^f)}})^2}}$  (8)

Ensemble mean innovation vector, which denotes the center of the mean $OmE$ distribution and should be close to zero:

$OmE_{bias} = \overline{\boldsymbol{d}}$  (9)

Standard deviation (i.e. 1std) of the debiased mean $OmE$ distribution, which should be larger than $\sigma_f$ and it is valid even if

$OmE_{bias}$ is not small: $DOmE = \sqrt{\overline{(\boldsymbol{d} - \overline{\overline{\boldsymbol{d}}})^2}}$  (10)

In addition, we calculate quantiles in data space to assess the ensemble median Q2(50%), the mid-spread Q1(25%)-Q3(75%) and the ensemble envelope Q0(1%)-Q4(99%).

### 3.2.1 SST L4 gap-free gridded observations

The use of high-resolution SST observations appears necessary when it comes to validate ensembles at eddy-resolving scales, such as in BISCAY36 at 1/36° resolution. The OSTIA SST L4 gap-free gridded dataset is chosen for the evaluation of Ens1 perturbing physics. OSTIA is a near-real-time daily-mean product of foundation SST free of diurnal variability. The model proxy for the foundation SST has been chosen to be the temperature interpolated at 10-meter depth. The dataset is reconstructed from merged multi-sensor in-situ and satellite observations for the Global Ocean, delivered on a rectangular grid at 0.05°×0.05° horizontal resolution.

Figure 6 shows examples of consistency metrics for different datasets and periods based on innovation samples. An ensemble of OSTIA SST, randomly perturbed using a Gaussian law assuming an observational error of 0.5 $°C$, is compared with Ens1 (Fig. 6a). The Ens3 ensemble exhibits an identical statistical behaviour, since there is no feedback onto physics from the ecosystem model. The calculations are carried out through the whole period of the seasonal-range ensemble. For the S6 ensemble in Fig. 6b, calculations are performed in April 2012.

Both model and data ensembles appear to be compatible with each other since vicinities overlap (Fig. 6a). Assimilating those observations, with those error estimates, with an ensemble or Bayesian filter would probably be well-posed, at least in data space, in the sense that the joint uncertainty vicinities associated with both sources of information appears to be nonzero. The statistical properties showing that the ensemble is consistent, are the model error estimate $\sigma_f$ being lower than the innovation spread DOmE, and the ensemble mean innovation vector OmE$_{bias}$ contained within the observational error interval $\pm 0.5$ °C, which are desired conditions in both cases (Fig. 6b). It can be seen that the $\sigma_f$ spread indicator slowly evolves in time, mainly increasing, consistently with the fact that the perturbation mechanism remains active throughout the period. However, there are occasions where the Ens1 spread is reduced and the ensemble envelope is being under-dispersive and sometimes biased with respect to the observational pdf. This is observed over a short period during the spring shoaling of the thermocline. This model overconfidence is likely associated with missing error re-stratification processes in our stochastic protocol, such as for example, the vertical subgrid scale physics which are not perturbed (e.g. the vertical eddy viscosity and diffusivity coefficients calculated in the turbulent closure scheme).

Another remark comparing the seasonal-range Ens1 and medium-range S6 ensembles, is that the SST model errors initially increase with similar rates (Fig. 6b), despite the fact that they have different number of members and initial conditions. The stochastic restarts in these two experiments are identical and therefore, the stochastic fields applied in the perturbed tendencies are alike. However, after a few days the model errors appear to increase with different rates, pertaining to the different ocean-atmosphere states during December for Ens1 and April for S6.





In Fig. 6b, we compare the Ens1 SST spread with the reference ensemble Ens0 (*Vervatis et al.*, 2016). Both stochastic approaches in Ens0 and Ens1 ensembles can be considered as variants of the perturbed tendency scheme. In case of the AR(1) processes we have the option to perturb several variables in Ens1, compared to the stochastic modelling of the wind forcing in Ens0. Therefore, the former yields a larger SST spread during winter and at the end of the run, with their

difference ranging at about $0.05\ °C$ to $0.1\ °C$. At the end of the run both ensembles become biased with respect to the observations (cf. *Vervatis et al.*, 2016 for Ens0 and Fig. 6b for Ens1), since are not constrained by DA. During spring in presence of a strong thermocline the spread is reduced in both ensembles, suggesting that there are missing error processes in both stochastic protocols (i.e. in addition to the wind for Ens0 and all variables perturbed in Ens1).

### 3.2.2    SLA L3 along-track observations

The data space consistency analysis for Ens1 SSH is carried out with the CMEMS SLA L3 along-track product in Table 2, assuming a Gaussian uncorrelated observational error of $0.05\ m$. The product includes processed data from several altimetry missions and in specific, for our period of interest from Envisat and Cryosat-2, delivered at $14\ km$ resolution for filtered and sub-sampled corrected data. Both data and model include tides. The model includes pressure forcing and therefore, an inverted barometer (IB) correction is applied to the model and observations. In order to calculate the SLA model equivalent,

we use the Mean Dynamic Topography (MDT) of the parent model IBI36 (Mounir Benkiran pers. comm.).

Figure 7a shows the distributions corresponding to averages of all SLA along track observations crossing the BISCAY36 domain, as well as the Ens1, Ens0 ensemble envelopes in data space. Both model ensembles appear to have more energy at the weekly timescale, occasionally at shorter timescales, and overall at the seasonal timescale compared to the observations. In addition, the steric cycle of the observations appears to be weak in terms of contribution to the sea level variability. The

minimum levels in both model and data are reached in mid-February, 2012, whilst for the rest of the simulation the SLA model equivalent shows larger variability than the data, most likely attributable to the prescribed open boundary conditions. A notable difference is observed between Ens1 and Ens0 ensembles, with the former being larger possibly because of the IB response to $SLP$ perturbations, as well as to the different stochastic approaches in the wind perturbations. Ens1 model uncertainties are comparable with those of the observational error, reaching a few centimetres in magnitude, whereas Ens0

shows an unrealistic model overconfidence.

As long as the center of distributions can be considered to be meaningful, a safe assumption in assimilation schemes where all errors are considered Gaussian, we find that the $OmE_{bias}$ stays in fair number of cases within the observational error interval (Fig. 7b). The most consistent configuration is in the Abyssal plain, where the spread is increased due to mesoscale decorrelation of eddies after spin-up. In contrast, the largest statistical biases are observed in coastal regions and in the

English Channel. This is also confirmed by box/whisker plots in several regions as a means to visualize both distributions and their consistency (Figs. 7c-e). There are many cases where the joint uncertainty vicinities associated with both sources of information is clearly nonzero. However, several instances of strong bias are evident in two of the regions, sometimes to the point that vicinities become disjoint, e.g. during the week between 06 to 12 March, 2012 (Figs. 7c-e). Disjoint vicinities will lead to ill-posed assimilation problems, meaning that a solution will be obtained since all analysis schemes are convex, but

the result will be meaningless.

When inconsistency is found, a general reason is that other error processes are active in the model in addition to the ones generated by the range of Ens1 perturbations. This is more apparent in the Ens0, since in this particular ensemble there are no perturbations in $SLP$. Both ensembles appear as being occasionally under-dispersive, notably in coastal areas in the English Channel and the Celtic Sea (Fig. 7b). In the English Channel, we expect error processes such as residual tidal error,

enhanced by the presence of hard-to-model local tidal fronts, and occasional Kelvin waves propagating along the coasts. In the case of the non-isostatic response to atmospheric pressure (e.g. the non-IB response), missing error processes can be also





present in ocean processes, as well as in the *SLP* perturbations themselves. Some error processes seem also to be missing from the range of perturbations which we apply in the English Channel, hinting for instance at the presence of high-frequency errors currently unaccounted for, although we cannot entirely attribute the misses to any given error process with the tools at hand. Another possible reason for the observed statistical inconsistency is the lack of altimetry observations,

verified by thinning techniques when decimating observations (not shown). Future wide-swath altimetry products (e.g. SWOT, https://swot.jpl.nasa.gov/) are likely to provide better coverage in coastal regions and stronger constraints on models overall.

### 3.2.3 Ocean Colour L4 gridded observations with gaps

Central to this work, is the consistency analysis of ecosystem ensembles perturbing physics and/or biogeochemistry, against

Ocean Colour (OC) products. OC technique exploits different radiation wavelengths and reflectances emerging from the sea surface, affected by phytoplankton and corresponding to different water types. In this study, we use surface total chlorophyll produced for the Global Ocean in the framework of the ESA Climate Change Initiative (CCI) programme, made available through CMEMS. This is a merged data records product from multiple sensors and ocean satellite passages provided in gridded format at $4\ km$ resolution. The OC L4 product is reconstructed from L3 reprocessed daily composites applying 8-

days temporal averaging to fill in missing data. The spatial coverage for the specific ESA-CCI OC L4 product has data gaps. The OC chlorophyll proxy in models is often taken as an average of chlorophyll over the top 10% of the euphotic layer (Isabelle Dadou pers. comm.). In our case, estimating an euphotic layer of approximately $50\ m$, we decided to tune the observation operator to return a model proxy as the mean value of the first $5\ m$ of the water column.

Figure 8 shows data space results of ecosystem ensembles and innovation statistics assuming an observational error of

$0.3\ mg/m^3$. Innovation statistics are calculated in log space applying an anamorphosis function to transform ecosystem lognormal distributions into Gaussian distributions. Among the seasonal-range ensembles the Ens3 exhibits the largest chlorophyll spread, with Ens2 being the least dispersive, as shown for the medium-range ensembles in section 3.1 (Fig. 5). In Fig. 8a, the Ens3 model-data samples appear to be marginally compatible with each other, since vicinities overlap only partially (also for Ens1-2 not shown, Ens3 being the best). There are also cases of disjoint pdfs with the most prominent in

late-March. During this period, observations show a strong phytoplankton bloom weakly present in all ensemble members (but with the correct phase), resulting in disjoint distributions. Subsequent less intense blooms just before and after this event are also evidenced in both model and data samples, and in those cases the pdfs overlap only partial. In the same line of thinking with the previous discussed SST and SLA networks, assimilating those OC observations and with those error estimates, would probably be marginally well-posed, in the sense that the joint probability associated with both sources of

information appears to be nonzero only at times. The most consistent configuration for Ens3 after the spin-up period appears to be May-June.

In Fig. 8b, we illustrate the innovation metrics defined in section 3.2. It is verified that all ecosystem model ensembles are under-dispersive during the first three months of the simulation. After this period dispersion slowly increases over time. Towards the end of the run chlorophyll uncertainties appear larger than $0.1\ mg/m^3$, which is small compared with the

chosen observational error, but not small with respect to the chlorophyll abundance at about $0.3\ mg/m^3$. In general, the CR underestimates chlorophyll abundance compared with OC data and subsequently leads to under-dispersive ecosystem model ensembles. An overview of the temporal evolution of the spread $\sigma_f$ resulting from different stochastic protocols, indicates that chlorophyll uncertainty variations are mainly controlled by physical processes and their errors. Biogeochemical processes and their uncertainties have a moderate impact on model errors, except during periods of phytoplankton blooms

where both components of the coupled system are important. In line with this, ecosystem model errors are increased more in





periods of higher biological productivity, than in periods of low biological productivity (e.g. comparison of Ens3 and S8 in Fig. 8b).

An acceptable result examining the model spread $\sigma_f$ for all ensembles against the innovation spread $DOmE$, is that the former is always contained within the latter (Fig. 8b). However, this is not sufficient for consistency to be verified. This is

because there is a constant bias of larger amplitude than the model dispersion, except near the end of the series. As a result, the $OmE$ metric most of the time is not contained within the observational error interval.

## 4        Potential impact on ensemble data assimilation

The aim of this section is to illustrate the potential impact of our ensemble-modelled uncertainties on data assimilation, by means of multivariate representers and EnKF-type incremental analyses. In all cases, we investigate the impact of

observations onto unobserved variables, such as other data types or subsurface variables. The calculations are conducted with SDAP (Sequoia Data Assimilation Platform, https://sourceforge.net/projects/sequoia-dap/), whose functions were expanded to be interfaced with the NEMO platform and its biogeochemical component PISCES (cf. Appendix C).

### 4.1        Ensemble-based single observation representers

As a first step, we calculate representers (correlations or influence functions) of single observations from our ensembles,

following *Bennett et al.* (1996). The representers are shown to be related to the estimate of the prior ECM $\boldsymbol{P}^f$, with prior state errors calculated by the representer matrix $\boldsymbol{H}\boldsymbol{P}^f\boldsymbol{H}^T$. In an ensemble-based context the prior ECM is approximated by $N$ samples in a decomposed form as $\boldsymbol{P}^f = \boldsymbol{S}^f\boldsymbol{S}^{f^T}$, where $\boldsymbol{S}^f = \frac{1}{\sqrt{N-1}}\prod_{j=1}^{N}\left(\boldsymbol{x}_j^f - \overline{\boldsymbol{x}^f}\right)$ the square root matrix of the error-subspace, with ensemble mean $\overline{\boldsymbol{x}^f} = \frac{1}{N}\sum_{j=1}^{N}\boldsymbol{x}_j^f$. The control vector $\boldsymbol{x}^f$ consists of the following variables $\{SSH, T, S, Chl\}$. Convolution with a localization function is applied to constrain spurious long-distance correlations resulting from the small

ensemble size. In the current setup, we use a Gaussian function with cut-off radius 3° and e-folding scale 0.2°.

In Fig. 9a-f, we illustrate examples of zero-lag representers of single observation OSTIA SST, for three different locations on May 07, 2012. The representers are calculated from 40 members of Ens1 (Fig. 9a-d), also from a 20-member subset of Ens1 (Fig. 9e) and finally from 40 members of Ens0 (Fig. 9f), and are shown as correlations between SST and itself, in addition to all other surface variables of the control vector. The correlation structures reveal differences between the Abyssal

plain and coastal areas, as well as between variables. On the English Channel and the south Armorican shelf, the filament-shaped structures for SST, SSS and surface chlorophyll are linked to near-shore features, such as river discharges (e.g. Loire river plume), mid-shelf thermal fronts and tidal fronts. The SSS pattern is dipolar, which could be explained by meridional plume migration (Fig. 9b). Spring bloom can be seen on the shelf, which might be confirmed by the negative correlation between SST and chlorophyll (Fig. 9d), for example the surface layers heating up during spring with plankton depletion

following a bloom.

The SST and SSH appear as decorrelated in the domain due to large-scale atmospheric forcings directly influencing SST in the spring season, as well as low-frequency mesoscale variability (Fig. 9c). Due to the mixed conditions on the inner shelf at that time of year, the SSH response is relatively large-scale as it is associated with barotropic processes at the scale of the external Rossby radius. In case of smaller ensembles, i.e. 20 members (Fig. 9e) or under-dispersive ensembles, i.e. SSH in

Ens0 (Fig. 9f), correlations due to partially-converged statistics increasingly contaminate the pattern of representers. Those patterns are however not very different from the ones of larger in size or dispersive ensembles, at least in their general shapes and signs, and mostly the range of positive-negative values is amplified. Similar results of rather broad and symmetrical structures are also found in case of SSH single observation representers (not shown).





Zero-lag representers of single observation OC data are calculated from Ens3. The most important finding is that correlations exhibit a different behavior in the open-ocean and in the shelves (Fig. 9g-i). Chlorophyll autocorrelation structures appear broad and symmetrical (with respect to the single-observation location) in the deep ocean, with scales dictated by the vertical velocity field of the underlying mesoscale quasigeostrophic features. The latter is in agreement with the ecosystem model

error regimes discussed in section 3.1 and depicted in Fig. 5 investigating the medium-range ensembles. On the shelves, chlorophyll correlations appear more dipolar in nature, when calculated for instance with respect to the SST and SSS fields, representing smaller scale local conditions. Filament-shaped structures of negative correlations between chlorophyll and SST are again verified for the specific Ens3 stochastic protocol, indicating that model errors in primary production are mainly controlled by model uncertainties in physical processes.

## 4.2    Incremental analysis

In this section, we carry out an EnKF-type analysis step for the first two members-001/002 of all ensembles Ens1, Ens2, Ens3, as well as for different subsets and seasons, using multiple observations together. The analysis step consists in calculating a linear combination of representers weighted by innovation. This is identical to the classic analysis step of a stochastic Ensemble Kalman Filter with one difference: observations are not perturbed. Using unperturbed observations

leads to the loss of statistical consistency for the second-order moments and forbids calculating innovation-based diagnostics; but legit EnKF increments for individual members would be harder to interpret physically because of the observation noise. So we decided to keep observations unperturbed in this section.

We calculate the Kalman gain matrix $K = ( \varrho \circ P^f H^T ) ( \varrho \circ H P^f H^T + R )^{-1}$ from an ensemble, and multiply it by the (member-dependent) innovation vector $d = y^o - H(x^f)$, where $y^o$ denotes the (member-independent) observation vector.

Thinning techniques are applied to high density observational networks, in order to reduce the computational time and optimize DA performance. The symbol $\circ$ denotes the Schur (element-wise) product of two matrices and $\varrho$ is the smooth Gaussian correlation function to perform localization (cf. previous section 4.1). In this study, we focus our investigation on the analysis increment $K \cdot d$.

### 4.2.1    Incremental analysis using SST L4

In Figs. 10a-b, we illustrate the SST correction on the first two members-001/002 on May 7, 2012, using the prior ECM from Ens1 40 members and assimilating the OSTIA SST L4 dataset. Both SST corrections reveal a large scale N-S pattern, associated with the fact that the model appears to be cold biased in that period with respect to the observations over the Irish shelf and the English Channel. In addition, the incremental analysis show corrections for mesoscale processes in the Abyssal plain and shelf processes near the river plumes. An interesting feature is the correction of the Bay of Biscay sub-gyre located

in the area 4°-6°W and 44°-46°N, confirmed by consistent increments in SSH, SSS and especially in surface total chlorophyll, hinting at sub-gyre scale changes in the vertical velocity (Figs. 10c-e). Increments of the opposite sign between SST and chlorophyll indicate that physical processes in the Bay of Biscay, such as tidal mixing, slope currents, river plumes and open-ocean mesoscale activity, play an important role for the biological productivity of the area.

In order to evaluate the impact of the Ens1 on the 3D T/S and chlorophyll model update, we examine the increment profiles

(no vertical localization) on two specific locations in the Abyssal plain and the Armorican shelf for the first member-001 (Fig. 11). The correction profiles reflect a fairly deep mixed layer in winter and a thinner mixed layer during spring, with chlorophyll changes at sub-surface layers hinting at ongoing bloom-related changes. Distinct vertical shifts in the increment are associated with the shallow thermocline depth during spring (~10-30 m) and with the depth of the euphotic layer (~40-50 m) influencing the sub-surface vertical corrections of chlorophyll. In the Abyssal plain, at depths greater than 1000 m the

vertical T/S corrections are possibly linked to the upper-ocean low-frequency mesoscale circulation, which affects the deep





vortex dynamics. This is an intriguing result showing that deep model errors can be controlled by ensemble-based DA methods, in which model ensembles are generated by perturbing surface variables and assimilating data of near surface ocean properties.

### 4.2.2 Incremental analysis using Ocean Colour L4

In Fig. 12, we highlight the fact that most of the total chlorophyll correction, as seen from the increments of the first member-001, arise from uncertainties in physics (i.e. Ens1 and Ens3), though ecosystem model uncertainties in most areas in the domain do enhance the effect of the physics. Moreover, in case where physics is not perturbed (i.e. Ens2) one can see mesoscale features in the correction (e.g. Abyssal plain), since the physics and hence the dynamics are the same for all members, to the contrary of Ens1 and Ens3. At a sub-surface level of 15 m depth, a signature of the ocean bottom relief is

observed in the correction fields, propagating in the perpendicular direction with the seabed features (resulting to a parallel crest-through signal) over the Celtic shelf near 7°-8°W and 48°-49°N. It is expected that biogeochemical processes will be sensitive to bottom Ekman pumping, especially in shallow shelf regions, and possibly to internal tides and waves induced by barotropic tides and winds, in this Bay of Biscay high-resolution configuration.

    An interesting remark is the opposite sign in increments, e.g. over the Celtic shelf, derived from the different model

ensembles of the coupled system, and in specific between the Ens1 perturbing physics and the Ens2 perturbing biogeochemistry. High positive increments over the shelves for all three ensembles, suggest that chlorophyll abundance is underestimated in those areas during spring subsequent blooms. The latter holds true in the continental shelf break near 3°-4°W and 46°-47°N, an area dynamically controlled by barotropic/baroclinic tidal processes, which in turns contribute to vertical mixing and enhance primary production. It is worth noting that in some coastal areas where the coupled system

appears to underestimate chlorophyll abundance, a very small correction is applied instead, due to the model overconfidence (i.e. under-dispersive ensembles) with respect to observations and their errors. Figure 10e and Fig. 12a present diverse correction patterns when different observations are assimilated, hinting at dissimilar processes captured by the prior ECMs in combination with each network, e.g. in these cases by SST or OC respectively. Following this latter argument comparing the increments of Fig. 10e and Fig. 12a, it is apparent that the chlorophyll correction is rather small when OC is not assimilated,

possibly because of weak cross-covariances between ocean and ecosystem properties.

### 4.2.3 Incremental analysis using both SST L4 and Ocean Colour L4

    In this section, we illustrate the multivariate impact of both temperature and total chlorophyll by using two observational networks simultaneously, namely the OSTIA SST L4 and the OC L4 (cf. Table 2). Changes to analyses with respect to univariate DA networks are observed for all variables varying from moderate to being locally large. In Figs. 13a-d, we depict

the correction fields on May 7, 2012 of the state vector surface variables for the first member-001, based on ensemble covariances of Ens3 40 members.

    The SST N-S correction pattern is not as distinct as the one when assimilating only OSTIA SST, especially in the English Channel area (Fig. 13a vs. Fig. 10a). In addition, the increments appear amplified, especially over the Irish shelf, with less refined patterns. The latter is also true for other surface variables, such as the SSH and SSS (Figs. 13b-c vs. Figs. 10c-d).

This effect is attributed to the presence of OC data in the solution of the DA convex scheme, in a situation where vicinities might be disjoint. In such a case, the analysis scheme possibly impacts more ecosystem properties than ocean physics. In the same line of thinking, chlorophyll correction values are moderately decreased when both observational networks are assimilated, compared to the analysis when assimilating only OC L4 (Fig. 13d vs. Fig. 12c). It should be noted that in case of chlorophyll, the increments appear less sensitive to changes when multivariate observing networks are brought together,

compared with the correction in physical properties.





In addition, we investigate the convergence of covariances and its impact on the increment analysis, incorporating different ensembles and ensemble sizes (i.e. Ens1 vs. Ens3 and 10 vs. 40 members; Figs. 13a, d, e-h). Modifications in the SST analyses between Ens3 and Ens1 40 members in a multivariate context are minor (Fig. 13a vs. Fig. 13e). If we use fewer ensemble members, the analyses for both ocean physics and biogeochemistry properties resemble the correction patterns of

larger ensembles. However, the increments are notably less smooth because covariances are calculated from partially converged statistics (Figs. 13e-f vs. Figs. 13g-h). A final remark is that in case of chlorophyll assimilation in conjunction with OSTIA SST, the analysis scheme moderately contributes to small scales, for all variables augmenting the increment values around ocean coherent dynamical features.

## 5      Discussion and conclusions

In this study, our contributions were specifically targeted at the generation of ensembles, in particular (but not solely) for high-resolution ocean configurations including regional and coastal physics and biogeochemistry. In addition, we sought to verify those ensembles against observational networks monitoring upper-ocean properties, in the sense of nonzero joint probabilities between model and data. As final step, we have illustrated the potential impact of those ensembles would have, once validated, on assimilated and unassimilated variables of the coupled system. The most important paradigm of this work

was to adopt a balanced approach building ocean-biogeochemical regional model ensembles and testing their relevance.

Our stochastic implementation is based on first-order autoregressive processes in the context of an SPPT scheme (*Brankart et al.,* 2015), applied to several sources of model uncertainties in the coupled system. These tendencies emerge from assumptions subject to erroneous atmospheric forcing, ocean model improper parameterizations and ecosystem model state uncertainties. The method is complemented to account for spatial correlations and anamorphosis transformation of

anisotropic uncertainty patterns, which is of vital importance in high-resolution regional configurations. The implementation is compatible with the enhanced MPI strategy of NEMO for double parallelization in the spatial domain and the ensemble dimensions (*Bessières et al.*, 2017).

Wind uncertainties are found to dominate all other atmosphere-ocean sources of model errors. The Ensemble spread, after a spin-up period of one month, focusing on upper-ocean properties is approximately 0.05 m for SSH and 0.5 °C for SST,

though these values vary depending on season and cross shelf regions. Ecosystem model uncertainties resulting from perturbations in physics appear to be larger than those perturbing the *SMS* concentration of the biogeochemical compartments, resulting in total chlorophyll spread slightly larger than 0.1 mg/m$^3$. The statistical spin-up period for biogeochemical variables appeared longer, $O(3$ months).

The validation of ensembles with respect to the gridded gap-free OSTIA SST L4, assuming an observational Gaussian error

of 0.5 °C standard deviation, suggests that the seasonal-range ensembles perturbing physics appear to be fairly consistent with the data distribution. The joint probability associated with both sources of information appeared to be always nonzero, enabling assimilation of that dataset. However, Ens1 (likewise Ens3 which is statistically identical with Ens1 for physical properties) was under-dispersive in SST and sometimes biased with respect to the observational pdf. Overall, pattern consistency through $OmE_{bias}$ and $DOmE$ metrics was found fairly good with SST data, especially for the large scales.

Analysing the consistency of ensembles with respect to the along-track SLA L3 CMEMS product (observational Gaussian error of 0.05 $m$ standard deviation), we could see the presence of strong biases between the model and along-track data distributions. All ensembles following various stochastic protocols for sea level (e.g. this study Ens1, likewise Ens3, and Ens0 following *Vervatis et al.*, 2016) appeared under-dispersive, notably in coastal regions. Consistency was improved for the open-ocean as a result of SLP perturbations with 5-day correlation timescale, having an isostatic effect triggering an IB

pumping on the Abyssal plain. The sea level model-data misfits were found to be associated with strong SSH spatial





gradients, in particular in the shelf regions such as the English Channel and Celtic Sea. Some error processes seemed to be missing from the range of perturbations which we applied in the English Channel – in particular, some high-frequency error processes are currently unaccounted for. Overall, we could not clearly attribute the missing processes to any particular error process with the tools at hand. Probabilistic "attribution" approaches are likely to provide more insight for sea level (e.g.

*Hannart et al.,* 2016).

Regarding chlorophyll, our consistency analysis in log-space showed a statistical biogeochemical spin-up time of $O(3$ months). During this spin-up period the ecosystem model ensembles Ens1, Ens2 and Ens3 appeared as being under-dispersive and biased with respect to gridded OC L4 data (log-transformed observational Gaussian error of $0.3\ mg/m^3$ standard deviation). The situation slowly improved over time for the second half of the simulation. In all three cases, the

model and observational ensembles appeared to be marginally compatible with each other. The most consistent configuration for chlorophyll appeared to be May-June for Ens-3. Statistical consistency was not always verified for chlorophyll as it was for SST and to a lesser extent for SSH. It seems difficult to attribute those error patterns to specific physical or biogeochemical processes, without further probabilistic "attribution" analysis.

Finally, we illustrated the impact of those ensemble-modelled uncertainties on data assimilation, by means of multivariate

representers and EnKF-type analyses in which observations were not perturbed, as a step towards developing an assimilation scheme. We first calculated correlations and representers of single observations, and in the second step we calculated analyses with multiple observations together. One objective was to access the impact of observations onto unobserved variables, such as other data types or subsurface variables.

The most important findings in the incremental analysis include the following. Corrections on physical properties are

associated with large-scale biases, notably a N-S pattern for SST between open-ocean and the shelves. Small scale local corrections are mainly visible over the shelves in near-shore coastal areas, explained by meridional river plume migration, mid-shelf thermal fronts and barotropic/baroclinic tidal processes. Incremental analysis on the water column structure denotes vertical changes linked to thermocline seasonal variability, such as for example the winter extended MLD and the spring shoaling of the thermocline. Distinct vertical shifts in the increment are also sought at depths near the euphotic layer

(~40-50 m) controlling the sub-surface vertical corrections of chlorophyll. The scales of the correction patterns in the Abyssal plain are often characteristic of the underlying quasigeostrophic mesoscale features, chlorophyll included due to the mesoscale vertical velocity field. Most of the chlorophyll correction arise from uncertainties in physics, but biogeochemical model errors tend to enhance the effect of the physics. Logically, assimilating chlorophyll seems to have a very measurable impact on physical variables.

Our ensembles were found on some occasions under-dispersive and additional approaches should be envisaged in future steps of this work, to augment model errors mainly for ecosystem variables. To list a few for the next phase (a) inflation techniques in the initial conditions may open some degrees of freedom in the first time-steps of the model run, (b) perturbing the biogeochemical parameters (see also *Garnier et al.,* 2016) in addition to *SMS* concentrations, activate the feedback of biology onto physics in the NEMO-PISCES coupled system, and (d) incorporating atmospheric ensembles, such as the

ECMWF-EPS system increasing the number of members to 50 (https://apps.ecmwf.int/archive-catalogue/).

One next step will also consist in analysing consistency patterns in data space between ensemble spread and ensemble of innovations. This can be most effectively achieved e.g. in the space spanned by array modes (*Le Hénaff et al.,* 2009; *Lamouroux et al.,* 2016; *Charria et al.,* 2016), which can be calculated with our ensembles (*Vervatis et al.,* in prep.).

In line with the above, the dynamics of regional (nested) models are largely controlled by the open boundary (OB)

conditions, with of course some dependency on the considered geographic area. Bay of Biscay dynamics is influenced by the North East Atlantic circulation, especially along the southern slope, with the seasonal reversal of the Iberian Poleward





Current (IPC). At depth, the entrance from the south of Mediterranean water masses has been shown to influence the Bay of Biscay hydrology between 600 and 1500 (*Koutsikopoulos and Le Cann*, 1996) and potentially the circulation (through interactions between eddies and deep salty lenses (see for instance *Carton et al.*, 2013).

Uncertainties on the OB conditions, either on the numerical scheme or on the prescribed values in case of active boundaries, are therefore expected to contribute significantly to the model error budget (see also *Kim et al.*, 2011). So it is natural to consider perturbing the OB conditions. This would very likely lead to an increase of the ensemble spread from the surface to at least 1500 m. However, the methods to perturb OB are not straightforward, in particular because of the need to ensure physical consistency between the perturbed variables, and because the errors on the prescribed fields at the OB are usually unknown. A favourable situation occurs when an ensemble of nesting or parent solutions is available and provides an estimate of OB uncertainties to the child model (*Ghantous et al.*, in review 2018).

## 6        Code and data availability

The ensemble simulations have been performed with the NEMO platform (http://www.nemo-ocean.eu/), in its stable version 3.6, freely available under the CeCILL public license. The code was downloaded from Mercator Ocean svn repository based on the NEMO revision 12956. The following cpp keys have been used to compile the code: key_top, key_pisces, key_bdy, key_tide, key_dynspg_ts, key_ldfslp, key_zdfgls, key_traldf_c2d, key_dynldf_c2d, key_vvl, key_mpp_mpi, key_iomput, key_xios2. The XML I/O Server is based on XIOS version 2.0 revision 1490.

The latest SDAP (SEQUOIA Data Assimilation Platform; De Mey- Frémaux, pers.comm., 2018) for UnixTM and Linux is freely available from the repository of the project's web page https://sourceforge.net/projects/sequoia-dap/. The SDAP system components are distributed under the GNU General Public License.

The additional algorithm described in Appendix A is also available from the open-source software platform Zenodo with doi:10.5281/zenodo.2556530. The FORTRAN subroutine integrated within the "stopar.F90" module is compatible with the NEMO MPI environment and it is delivered without any warranty, declining any responsibility for errors, or improper usage.

The model ensemble output, the observations and the tools to process data are accessible upon request at the ECMWF premises, through the DHS (Data Handling System) and the ECFS (ECMWF's File Storage) client-server application.

**Acknowledgments.** This work was carried out as part of the Copernicus Marine Environment Monitoring Service (CMEMS) "Stochastic Coastal/Regional Uncertainty Modelling (SCRUM)" project. CMEMS is implemented by Mercator Ocean in the framework of a delegation agreement with the European Union. Acknowledgement is made for the use of ECMWF's computing and archive facilities in this research. This work was also supported by computational time granted from the Greek Research & Technology Network (GRNET) in the National HPC facility – ARIS – under project ID PA002007.

**Appendix A:** Elliptic Gaussian equation in NEMO MPI environment

The stochastic parameterizations in this paper are based on the ensemble capabilities of NEMO (*Brankart et al.*, 2015; *Bessières et al.*, 2017). We use the generic FORTRAN codes included in the trunk NEMOGCM/NEMO/OPA_SRC/STO/. Our work complements NEMO stochastic modules in terms of explicit calculation of AR spatial scales, by solving an elliptic Gaussian equation. A subroutine called "sto_par_xygau" is integrated in "stopar.F90" module. Various options remain valid, such as the use of Laplacian filtering, anamorphosis functions, higher order AR processes for both SPPT and SPUF methods. The programming flowchart illustrates the integrated algorithm in NEMO MPI environment.





| **Algorithm** sto_par_xygau | |
|---|---|
| Purpose: | Solve the elliptic Gaussian equation $\sum_{i=1}^{m} \mathcal{G}(\boldsymbol{r})^{(i)} = 0$. |
| Method: | Work on MPI communicator allocated per ensemble member. |
| | On zero-processor create the vectors of the centers $\overline{\boldsymbol{r}} \in \{\overline{r_1}, \dots, \overline{r_m}\}$ and the variances $\boldsymbol{\sigma_r}^2 \in \{\sigma_{r_{11}}{}^2, \dots, \sigma_{r_{mm}}{}^2\}$ of the Gaussian distributions. The centers are randomly selected on a few model grid points. The length scales vary randomly in the two dimensions around a typical correlation length per tendency. |
| | Broadcast the vectors on the rest of the processors for all sub-domains (call MPI_BARRIER, call MPI_BCAST). |
| | Draw multi-modal spatial maps of $\mathcal{N}(0,1)$ distribution covering the whole model domain, by solving the elliptic Gaussian equation on a few model grid points $(i)$: $\xi^{(i)} \leftarrow \mathcal{G}(\boldsymbol{r})^{(i)}$. |
| | Normalize the stochastic amplitude by a factor of $f^{(i)}$ and global online diagnostics of statistical moments $\mu, \sigma$ to ensure $\mathcal{N}(0,1)$ distribution (call MPI_ALLREDUCE, call MPI_SUM): $\xi^{(i)} \leftarrow \{f^{(i)}, \mu, \sigma\} \cdot \xi^{(i)}$. |

**Appendix B:** Computational resources and performance

For a detailed analysis of the computational resources and performance for the generation of ensembles with the BISCAY36 configuration, the reader is referred to ECMWF's "Progress and Technical Reports" in https://www.ecmwf.int/en/research/special-projects/spgrverv-2016/.

We briefly recapitulate the most recent setup compiling and running the code at ECMWF HPCF. The model ensembles are carried out on CCA and CCB clusters, which are Cray XC40 systems integrating Intel Broadwell nodes, with 36 cores per node and 128 GB (2400 MHz DDR4) memory per node. The code is compiled under the Intel Broadwell software environment using the Cray Development Toolkit (CDT) cdt/17.03, with intel/17.0.3.053 compiler, and the following libraries: cray-netcdf-hdf5parallel/4.4.1.1 and cray-hdf5-parallel/1.10.0.1. The same environment is used for the compilation

of XIOS version 2.0. We use -O3 optimization in the FCFLAGS of the compilation architecture file. The model output consists of daily files of the ocean state vector and the two classes of chlorophyll, as well as three-day averages of 14 3D-biogeochemical variables.

BISCAY36 scales-out using 96 processors of domain decomposition per ensemble member, excluding land processors. Taking under account the ECMWF's hardware/software specifications, we have tested the following resources geometry: (a)

for 10 members, we have used 960 NEMO processors and 48 XIOS servers filling a total of 28 nodes, (b) for 20 members, we have used 1920 NEMO processors and 24 XIOS servers filling a total of 54 nodes. The ensemble simulations were submitted as batch jobs for a 30-day run. For these examples, the ECMWF's job epilogue during production returned information for the runtime average at about 489 minutes, with runtime standard deviation at approximately 29 minutes, including the first/last reading/writing time-steps.

**Appendix C:** SDAP and NEMO-v3.6/PISES-v2 interface

Within the CMEMS Service Evolution project SCRUM (Stochastic Coastal/Regional Uncertainty Modelling), an ensemble-based consistency analysis toolbox (nicknamed "scrumcat": SCRUM analysis toolbox) has been developed to document the statistical consistency of ensemble-based model uncertainties with respect to observations and their errors. The "scrumcat" toolbox is built upon the Sequoia Data Assimilation Platform (SDAP; https://sourceforge.net/projects/sequoia-dap/; De Mey-

Frémaux, pers.comm., 2018) and offers the following advantages:

- Fully modular
- Provides all data structures and services required by an ensemble-based application
- Interfaced with the European FP7 SANGOMA project Fortran library



- Written in Fortran-90
- Open Source licensing (GPL/CeCill)
- Versioned with svn
- SDAP Makefile and configuration tool
- Model grid services, including e.g. interpolation and regional masks
- Data services, including e.g. observation operators and decimation
- Off-Line Analysis services (OLA) and OLA explorer post-processing tool
- Interfaced with ocean modelling platforms; the most recent work includes NEMO-v3.6 and PISCES-v2 grid bathymetry, ocean-biogeochemical state variables in the control vector, observation operators
- NetCDF I/O: CMEMS data input, ensemble input, grid input, regional masks input
- An observational error covariance model
- Array-space consistency diagnostics
- Runs on several platforms, including ECMWF HPC machines under intel, gnu and cray programming environments

SDAP is a recent (2014) rewrite of Sequoia, a modular assimilation system builder developed by P. De Mey with post-docs

and LEGOS engineers, and disseminated via the SIROCCO French coastal ocean modelling national service (http://sirocco.omp.obs-mip.fr/) and SANGOMA European project (http://www.data-assimilation.net/). Over the years, Sequoia and SDAP have been used in research, operational forecasting, and for industrial partnerships.

SDAP can work with structured grids (finite difference) as well as with finite elements/finite volume grids. Its standardized interface with numerical models allows it to couple with virtually any model. In-memory coupling with numerical models is

supported. The user must provide a number of support routines with a predetermined interface and extend the model code with a few calls to SDAP routines.

The system of interchangeable analysis kernels allows using several assimilation algebra, among which a full-rank kernel solved in the dual space, used in a 4D localized implementation of the EnKF (beluga) using MPI for scheduling of parallel runs in an ensemble.

In its latest instance (v1.6), SDAP compiles under Intel, GNU and Cray FORTRAN compilers, and runs on desktop and laptop PCs, Intel Xeon clusters, and supercomputers such as Cray HPC.

Ongoing work with SDAP follows the double parallelization implementation of NEMO ensemble capabilities (*Brankart et al.*, 2015; *Bessières et al.*, 2017), both in ensemble domain and domain decomposition, aiming at:

- running ensemble methods (EnKF, EnOI, LETKF, Particle Filters) with MPI while the model already uses MPI domain decomposition
- using a correction and restart strategy maintaining the assimilation interface code within NEMO, by interfacing the ASM and OBS modules
- running ensemble methods efficiently on modern, 16+ core processors
- making ensemble methods compatible with pnetcdf (parallel implementation of NetCDF) and nVidia GPU-based
libraries (cublas and culapack)

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

**Table 1.** Table of simulations: Control Run (CR), ensemble medium-range sensitivity experiments (S1-8) and seasonal-range ensembles (Ens1-3). The table shows the stochastic parameterizations based on first order autoregressive processes AR(1). An elliptic Gaussian equation is applied to introduce spatial correlations with variable/anisotropic length scales (1std: $\sigma_r$ in degrees), temporal correlations ($\tau$ in days) and uncertainty amplitude (1std: $\sigma$ no units) of the 2D normal distributions (prior to anamorphosis function if applied). The spatial correlation value for the bottom drag coefficient $c_b$ is an approximation after 100 passes of the Laplacian operator. Ens0 is a seasonal-range ensemble performing stochastic modelling of the wind forcing based on EOF modes (*Vervatis et al.*, 2016).

| experiment | perturbed variables | uncertainty amplitude ($\sigma$) | correlation timescales ($\tau$) | spatial scales ($\sigma_r$) | distribution |
|---|---|---|---|---|---|
| CR | one year unperturbed free run (July 2011–June 2012) | | | | |
| Ens0 | wind perturbations based on EOF modes (*Vervatis et al.*, 2016) | | | | |
| | | atmospheric forcing | | | |
| S1 | $\mathcal{U}$ | 0.3 | 3 days | 1° | Gaussian |
| S2 | $T_{air}$ | 0.1 | 15 days | 2° | Gaussian |
| S3 | SLP | 0.01 | 5 days | 3° | Gaussian |
| | | ocean model parameterizations | | | |
| S4 | $c_d, c_e, c_h$ | 0.1 | 3 days | 0.5° | Gaussian |
| S5 | $c_b$ | 0.2 | 30 days | 0.2° | Laplace flt* |
| | | synthesis of ocean-atmosphere model uncertainties | | | |
| S6 | S1-5 | medium range ens (April 2012; 20 mem) | | | |
| Ens1 | S6 | seasonal range ens (December 2011–June 2012; 40 mem) | | | |
| | | ecosystem state | | | |
| | | ocean-atmosphere state identical to the CR for all members | | | |
| | | 0.6 | 10 days | 0.5° | Lognormal** |
| S7 | SMS(C) | medium range ens (April 2012; 20 mem) | | | |
| Ens2 | S7 | seasonal range ens (December 2011–June 2012; 40 mem) | | | |
| | | synthesis of coupled ocean-biogeochemical model uncertainties | | | |
| S8 | S6-7 | medium range ens (April 2012; 20 mem) | | | |
| Ens3 | Ens1-2 | seasonal range ens (December 2011–June 2012; 40 mem) | | | |

abbreviations: flt-filter; ens-ensemble; mem-members
* 100 passes of the Laplacian filter in a Gaussian distribution per model grid point
**A lognormal anamorphosis function is applied in the *SMS* concentrations of the 24 PISCES prognostic variables $C$

**Table 2.** CMEMS observation product identifiers (http://marine.copernicus.eu/).

| Obs1 (daily freq.) | Product Identifier | Error |
|---|---|---|
| a) gridded 0.05$^o$ | SST_GLO_SST_L4_NRT_OBS_010_001 | 0.5$^o C$ |
| b) along track 14 $km$ | SEALEVEL_GLO_PHY_L3_REP_OBS_008_045 | 0.05 $m$ |
| Obs2 (8-days freq.) | Product Identifier | |
| gridded 4 $km$ | OCEANCOLOUR_GLO_CHL_L4_REP_OBS_009_093 | 0.3$mg/m^3$ |





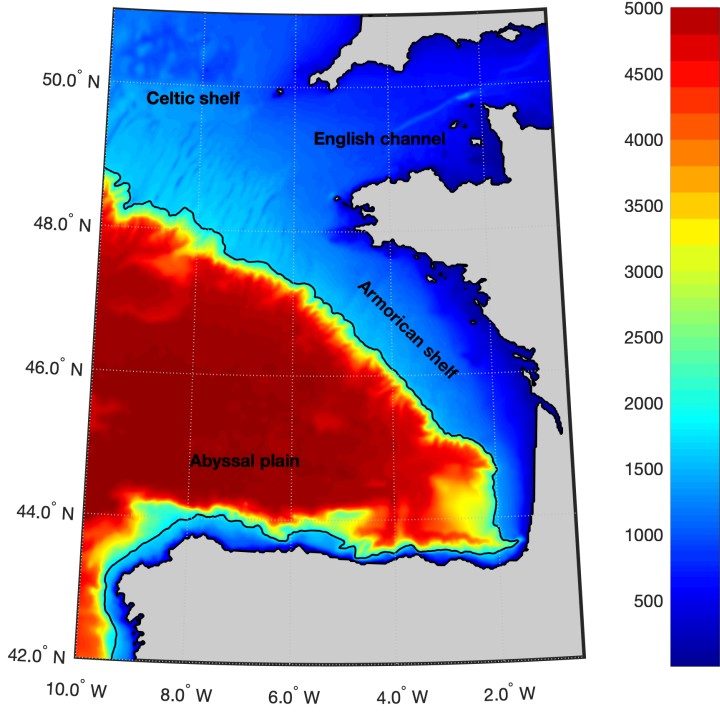

**Figure 1**      BISCAY36 model domain and bathymetry in meters. Black line denotes the 200 m isobath.





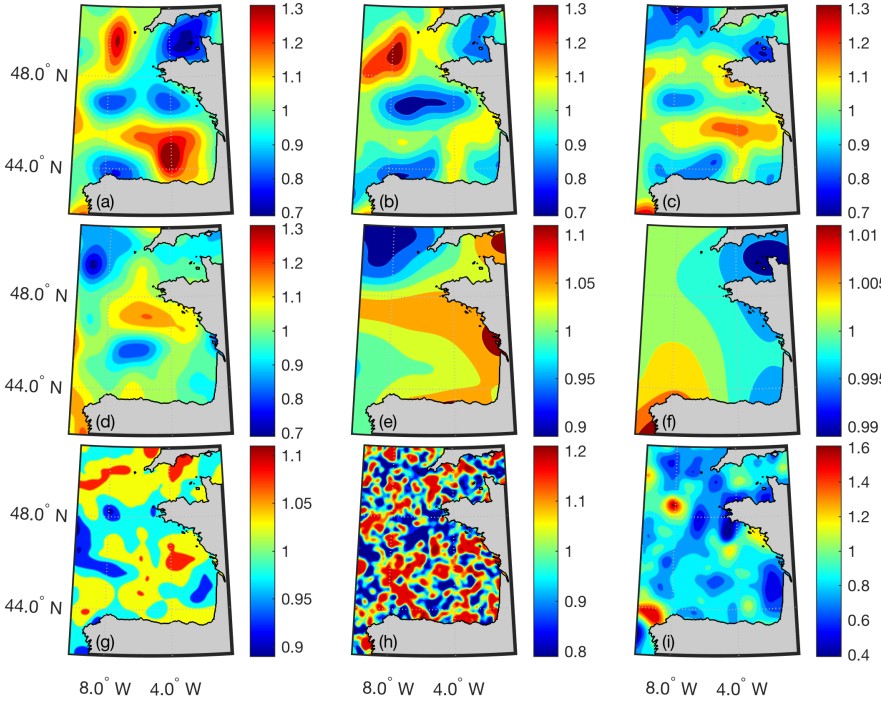

**Figure 2**   First-order autoregressive stochastic patterns, solving an elliptic Gaussian equation, applied in the perturbed tendencies schemes, see Eqs. (5-7): (a) wind $\mathcal{U}$ initial stochastic pattern of member-001; all other subplots as in (a) for: (b-c) the second and third day respectively, (d) member-002, (e) air temperature $T_{air}$, (f) sea level pressure SLP, (g) wind drag $c_d$ and turbulent coefficients $c_e$, $c_h$, (h) bottom drag coefficient $c_b$ applying a Laplacian filter, (i) lognormal distribution of Sources Minus Sinks $SMS(C)$ of biogeochemical tracers. Uncertainty amplitudes in colorbar and spatiotemporal scales are denoted in Table 1.





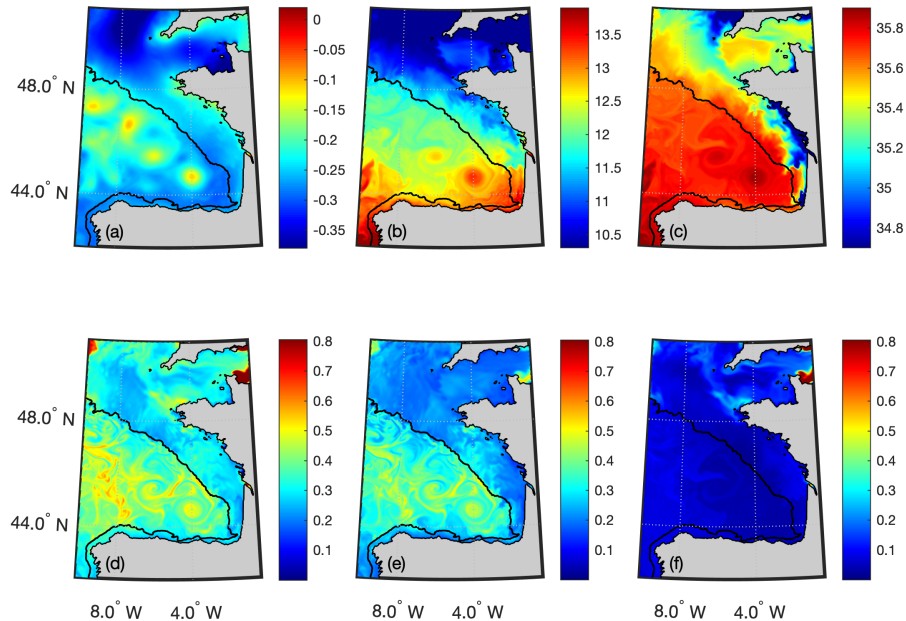

**Figure 3**        Control Run (CR) ocean model surface variables on April 30, 2012: (a) SSH in meters, (b) SST in °C, (c) SSS, (d-f) from left to right: total surface chlorophyll and the two classes "nano" and "diatoms" in mg/m$^3$. Black line denotes the 200 m isobath.





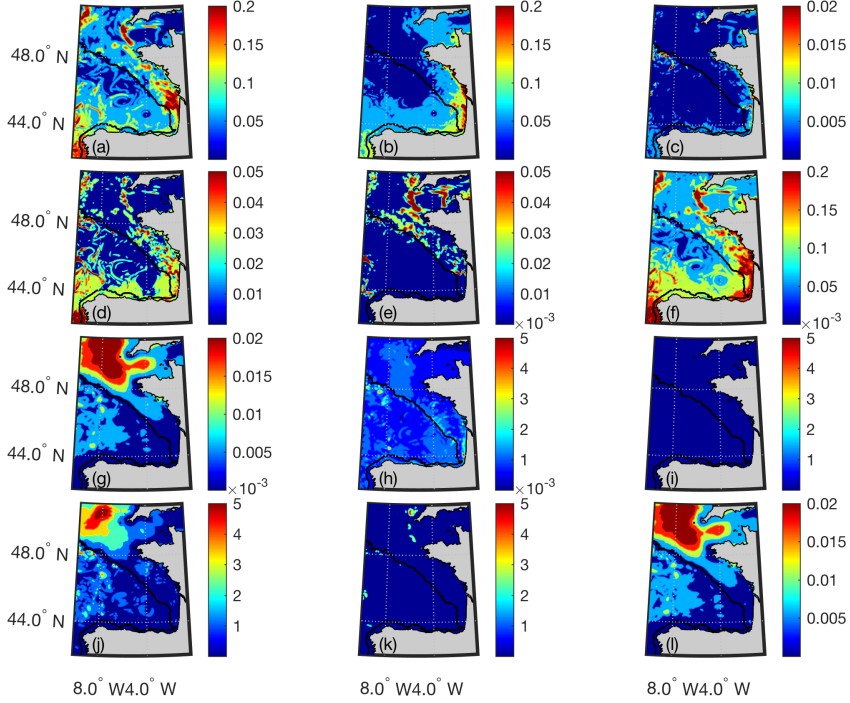

**Figure 4**        Model uncertainties of medium-range ensembles in Table 1, after one-month spin-up on April 30, 2012. SST spread (i.e. 1std) in °C for S1-6 experiments perturbing the (a) wind $\mathcal{U}$, (b) air temperature $T_{air}$, (c) sea level pressure SLP, (d) wind drag $c_d$ and turbulent coefficients $c_e$, $c_h$, (e) bottom drag coefficient $c_b$ (f) all variables together; (g-l) SSH spread in meters perturbing the same variables as in (a-f). Note the different colorbars in SST with units varying up to 0.2 °C and in SSH up to 0.02 m. Black line denotes the 200 m isobath.





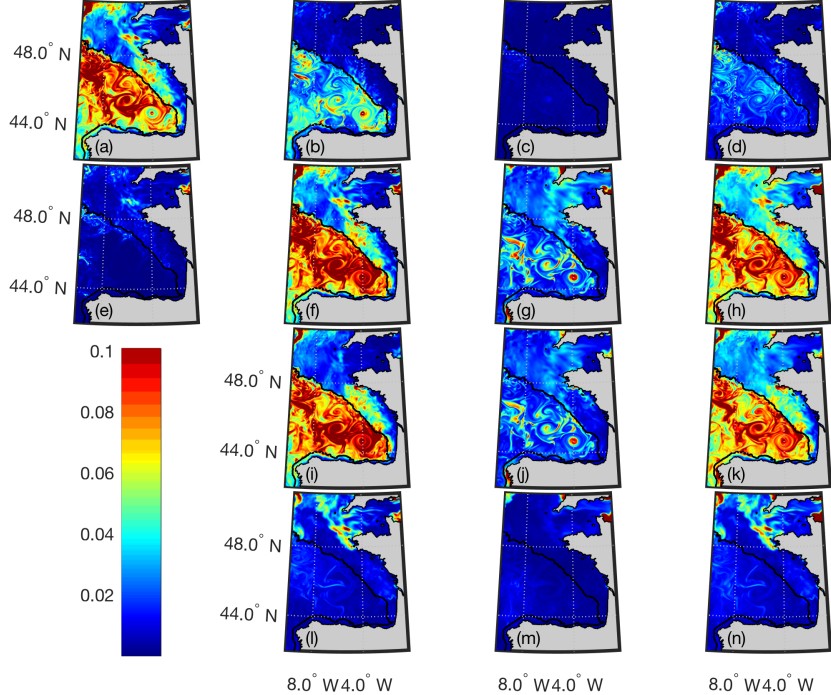

**Figure 5**    (a-f) model errors of S1-6 experiments for total surface chlorophyll concentration in mg/m$^3$, perturbing the physical variables described in **Figs. 4a-f**, (g) total surface chlorophyll spread in S7 experiment perturbing the sources minus sinks $SMS(C)$ of biogeochemical tracers, (h) total surface chlorophyll spread in S8 experiment perturbing all physical variables and $SMS(C)$ of biogeochemical tracers, (i-k and l-n) same as in (f-h) for nano-chlorophyll and diatoms spread, respectively. Black line denotes the 200 m isobath.





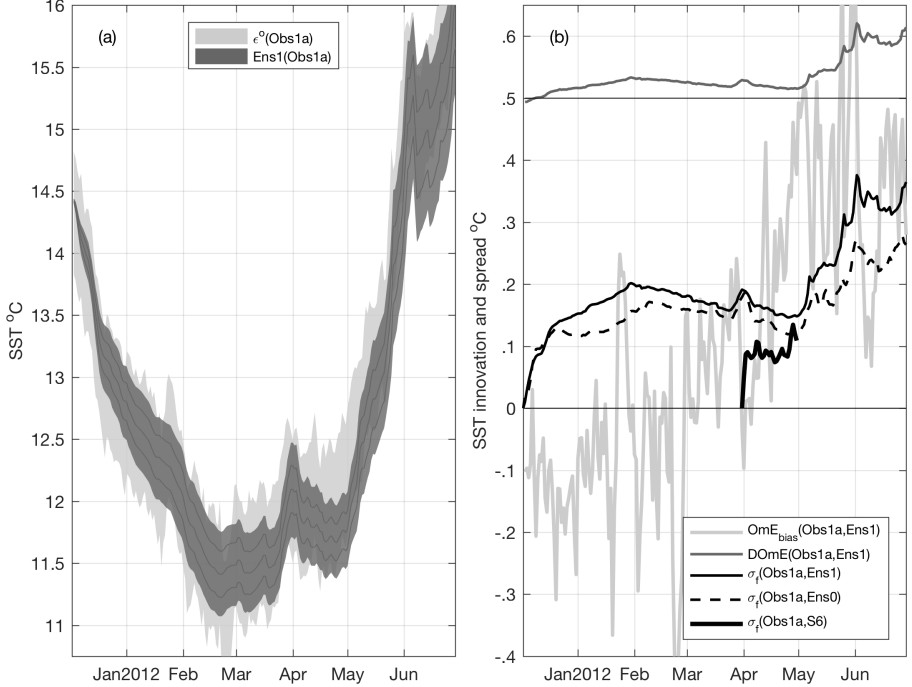

**Figure 6** (a) OSTIA SST L4 in °C observation distribution and Ens1 ensemble envelope/inter-quantile ranges in data-space, (b) innovation statistics and spread; thin horizontal line denotes the observational error 0.5 $°C$. Legend names declared in Tables 1, 2.



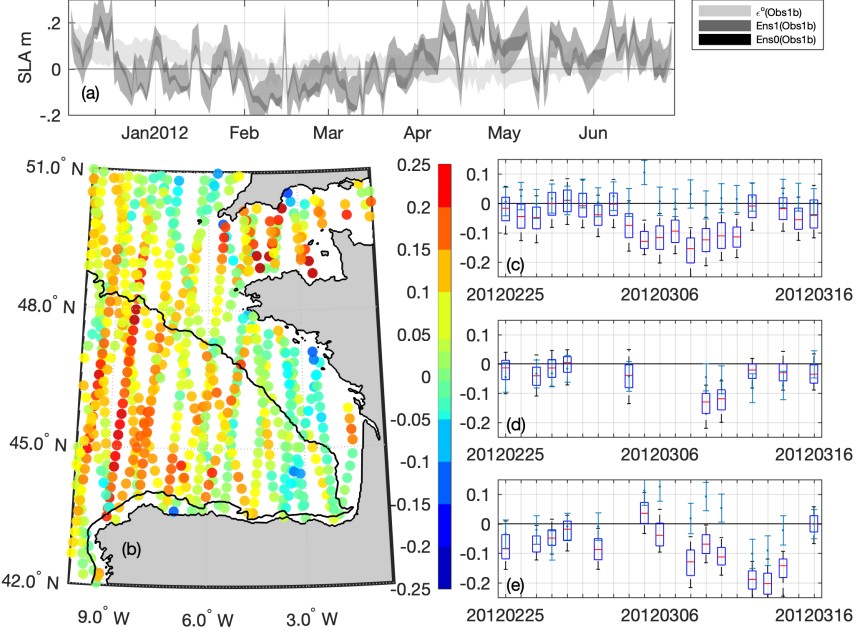

**Figure 7**   (a) SLA along track L3 observation distribution and Ens1, Ens0 ensemble envelopes in data-space (units: in meters; legend names declared in Tables 1, 2), (b) $OmE_{bias}$ map using Ens1 for the period starting on February 25, 2012 and for three consecutive weeks, (c-e) Ens1 box-whisker plots and observation error bars for the Abyssal plain, the Armorican shelf and the English channel, respectively. Black line denotes the 200 m isobath.





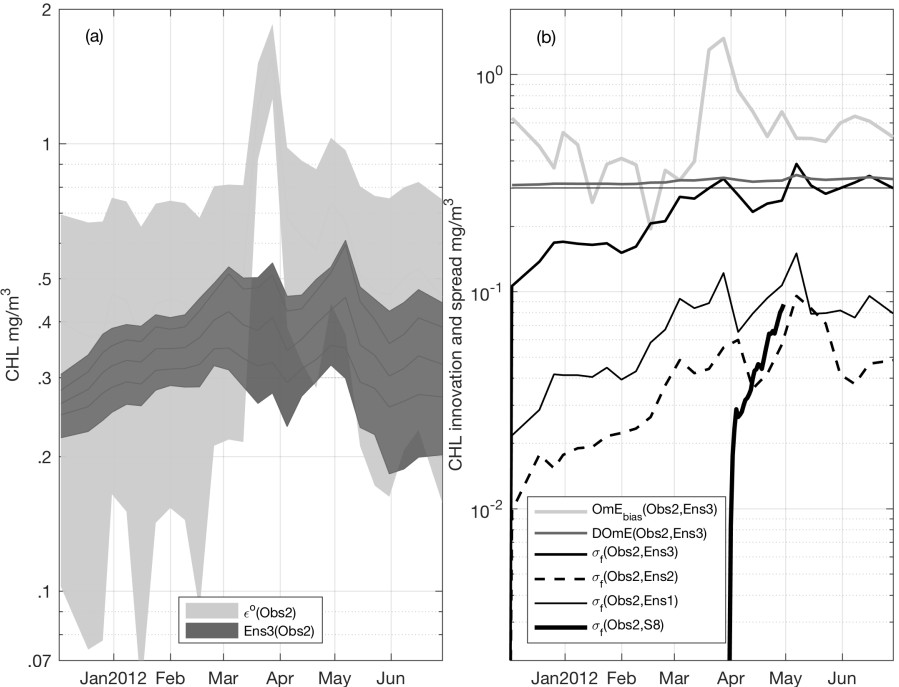

**Figure 8**    Same as **Fig. 6** for Ocean Colour L4 total surface chlorophyll observations and ecosystem model ensembles, with innovation statistics calculated in log space. The observational error is set at $0.3\ mg/m^3$. Legend names declared in Tables 1, 2.





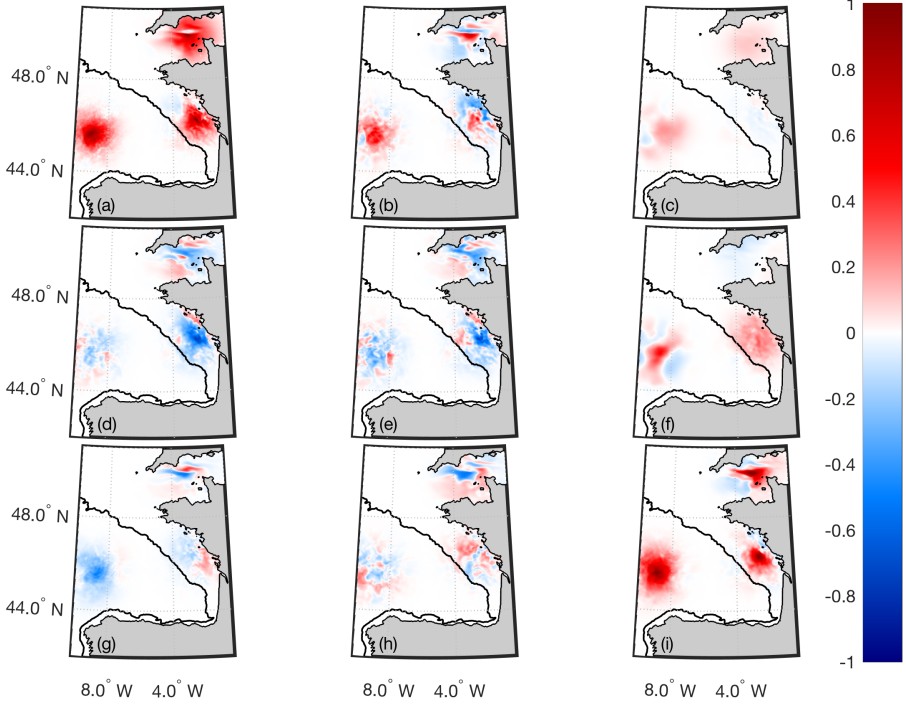

**Figure 9**         Zero-lag single observation representers for three different surface locations in the Abyssal plain, the Armorican shelf and the English channel, calculated as correlations between OSTIA SST L4 and all surface variables in the control vector, derived from 40 members of Ens1 ensemble on May 07, 2012: (a) cor(SST,SST), (b) cor(SST,SSS), (c) cor(SST,SSH), (d) cor(SST,CHL), (e) same as (d) for 20 members, (f) same as (c) for Ens0. Correlations between OC L4 chlorophyll observations and surface variables in Ens3 control vector: (g) cor(CHL,SST), (h) cor(CHL,SSS) and (i) cor(CHL,CHL). A localization Gaussian function is applied to suppress distant spurious correlations, with radius 3° and e-folding scale 0.2°. Black line denotes the 200 m isobath. For SST and OC CMEMS observational products cf. Table 2.





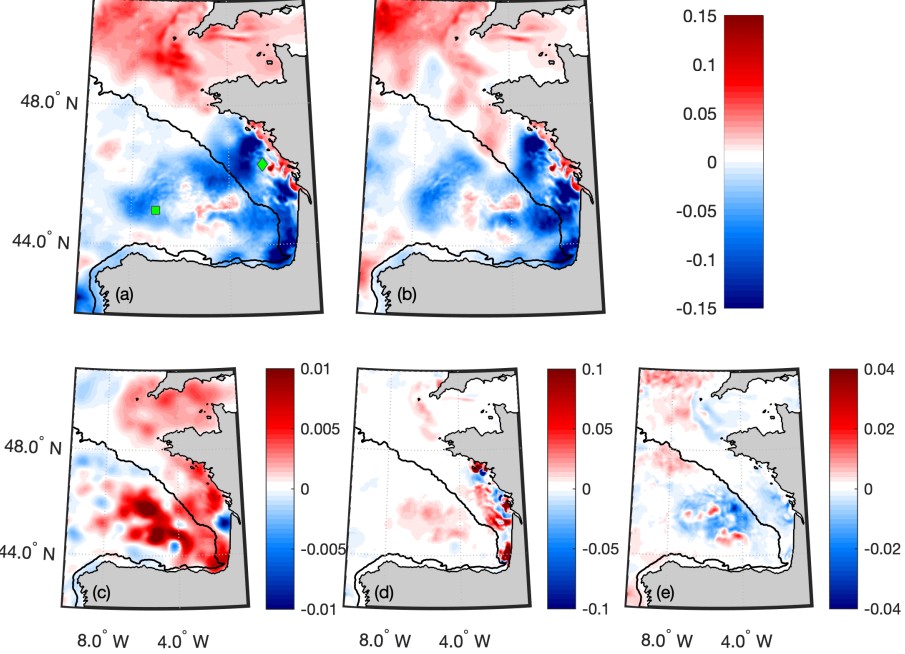

**Figure 10**        Incremental analysis using OSTIA SST L4 (cf. Table 2) on May 7, 2012 using Ens1 40 members: (a-b) correction on SST in °C for the first two members-001/002, (c-e) correction of the first member on SSH in meters, on SSS and on surface total chlorophyll in $mg/m^3$; (a) two locations illustrated in the following **Fig. 11** for the Abyssal plain (green square [7°W 45°N]) and the Armorican shelf (green rhombus [2.6°W 46.3°N]). Black line denotes the 200 m isobath.





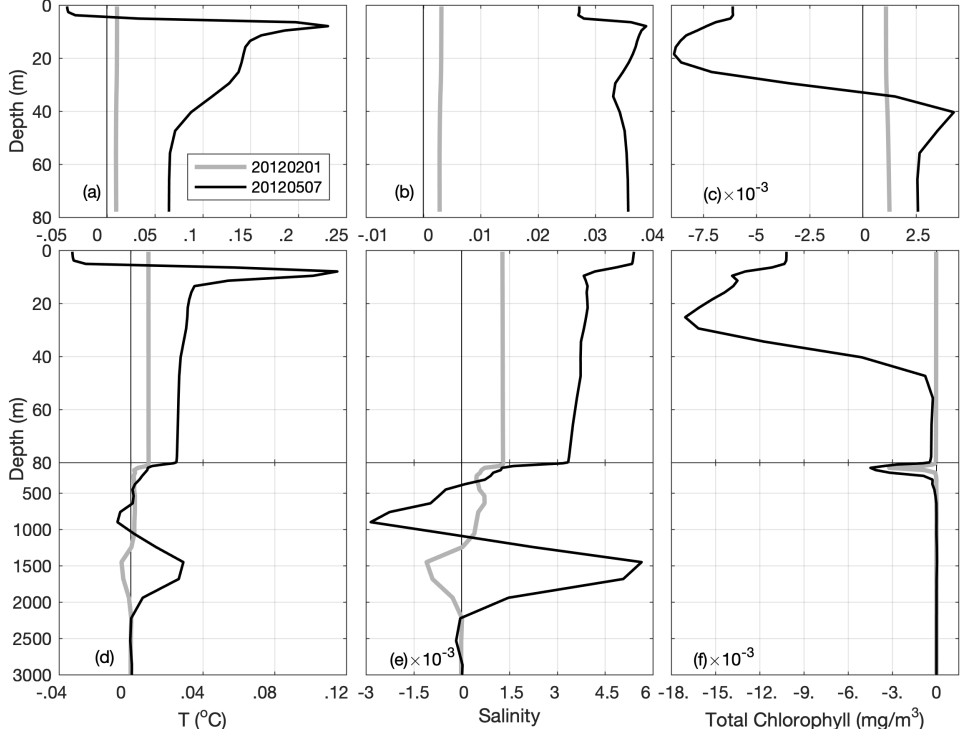

**Figure 11**       Incremental analysis using OSTIA SST L4 (cf. Table 2) on February 1, 2012 (grey lines) and on May 7, 2012 (black lines), using Ens1 40 members; cf. **Fig. 10a** for the two locations: (a-c) vertical correction of the first member-001 on T in °C, salinity and total chlorophyll in $mg/m^3$ in the Armorican shelf, and (d-f) same for the Abyssal plain.



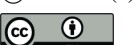

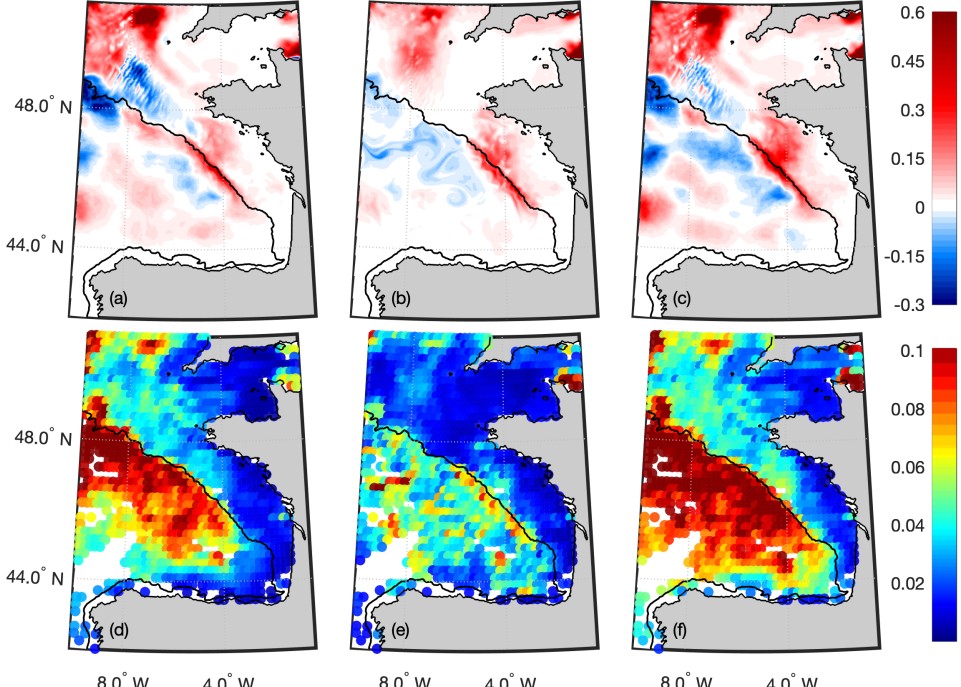

**Figure 12**         Incremental analysis using Ocean Colour L4 (cf. Table 2) on May 7, 2012 using ensemble covariances from 40 members and three different ensembles Ens1, Ens2, Ens3 (left to right in panels): (a-c) correction of the first member-001 on total chlorophyll in $mg/m^3$ at the sub-surface depth of 15 m, (d-f) prior model ensemble spread in data space of total chlorophyll in $mg/m^3$, as a mean value of the first 5 m of the water column. Black line denotes the 200 m isobath.





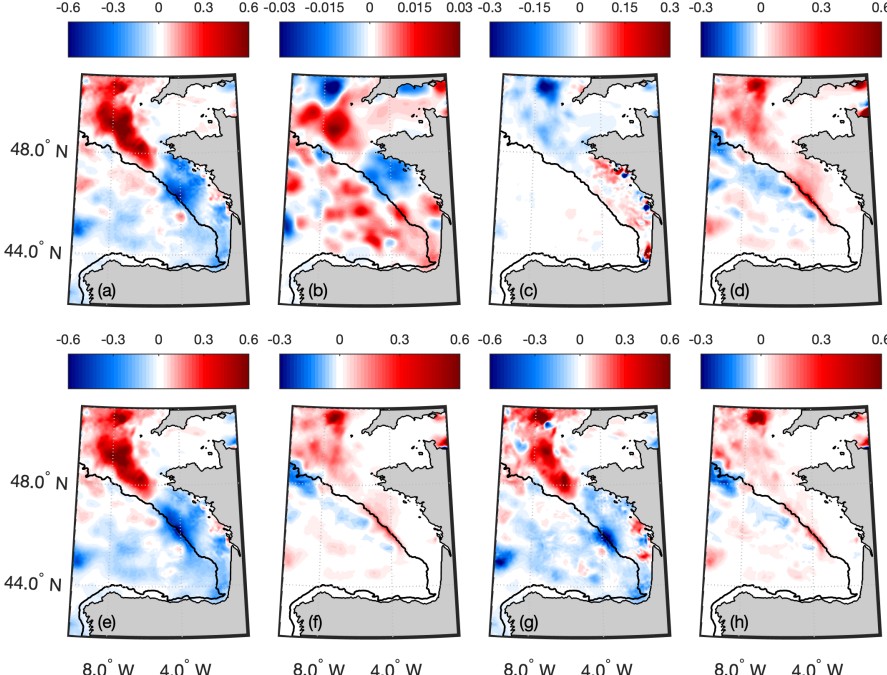

**Figure 13** Incremental analysis using OSTIA SST L4 and Ocean Colour L4 (cf. Table 2) on May 7, 2012: (a-d) ensemble covariances are calculated from Ens3 40 members; correction of the first member-001 (from left to right) on SST in °C, on SSH in meters, on SSS and on surface total chlorophyll in $mg/m^3$, (e-f) same as (a) and (d) calculated from Ens1 40 members, (g-h) same as (e-f) calculated from Ens1 10 members. Black line denotes the 200 m isobath.