# Peer review of "Physical-biogeochemical regional ocean model uncertainties stemming from stochastic parameterizations and potential impact on data assimilation"

_Geoscientific Model Development, 2019_

## Referee Comment (RC1) · Anonymous Referee #1 · 26 Apr 2019

The general purpose of this paper is to produce ensemble simulations with a physical-biogeochemical model of the Bay of Biscay, to assess these simulations using sea surface temperature, sea level anomaly and ocean color observations, and to evaluate the potential impact of the new ensembles in a Kalman data assimilation framework. In terms of model development, the novelty is in the introduction of long-range horizontal correlations in the stochastic parameterization of the NEMO ocean model. In terms of application, many ensemble simulations are performed to evaluate the effect of a variety of stochastic perturbations. In my view, the general approach is meaningful,

even if the paper is clearly more directed towards practical application than model development.

However, for the reasons explained below, I struggled a lot to read the paper and to write a review, mainly because there are so many different things involved (as acknowledged in the introduction: "the scientific objectives cover a broad spectrum of interdisciplinary components"), and because the text is often unclear and makes it difficult to figure out what is effectively done (despite the many details provided). I also found that the methods used to simulate the large-scale perturbations and to evaluate the results are questionable, without sufficient explanations and justifications. Overall, I do not know if these problems can be solved or not, but this should certainly require a thorough reconsideration of the text to improve the clarity of the arguments (more simple, more factual), and to provide the missing explanations and justifications.

**Main comments**

1) I do not understand the method used to generate the large-scale perturbations. I do not know what an "elliptic Gaussian equation" is. I checked the expression in google, and I had only 7 replies, all of them (except one) from the author's work. A classic approach to generate a long-range correlated noise from a white noise is to solve an elliptic partial differential equation like $E(x) = w$, where w is a white noise, x the resulting correlated noise, E an elliptic operator, like a Laplacian operator (times the square of a length scale). I checked the code provided with the manuscript and I found no solver for an elliptic partial differential equation. Anyway, whatever the method, we need to understand what it is, how it works, what are the benefits with respect to other methods, and what are the expected properties of the resulting noise in terms of correlation structure. If not, there is no real model development in this paper, and it is probably better to move to another journal.

2) I do not really understand the method that is used to compare the stochastic ensemble simulations to observations. This is at the core of the paper and would need

to be based on a solid ground. The results of the comparisons are displayed inf Figs. 6, 7 and 8, which mainly show averages or mixtures over the whole domain. On the one hand, what is done to compute these statistics is unclear to me, and it is difficult to have a clear idea of what stands behind the various curves that are shown. On the other hand, I do not understand how these global results can be used to deduce that the ensemble are more or less consistent with the observations. It is indeed repeatedly stated in the paper that the system is very heterogeneous, and I thus presume that the local consistency with observations is an important issue. In summary, we need to understand the logic supporting the validation procedure, and why it is applicable to this problem.

3) The paper is made of three different components (description and development of the method, evaluation of the ensemble simulations, and impact on data assimilation), which are presented and discussed almost independently. I understand that correlated stochastic perturbations were needed to produce the ensemble, and that good ensembles are likely to improve data assimilation. I think this is not sufficient to connect the components of the paper together. For instance, the effect of the new methodological development on the ensemble simulations is not specifically discussed, and the effect of data assimilation is mainly described in terms of realism and physical consistency than in relation to what is done in the rest of the paper.

**Other comments**

1) A previous paper by Vervatis et al. (2016) is cited throughout the paper. I think it would be necessary to better position the present paper with respect to the previous one.

2) The appropriate reference for the SPPT scheme is Buizza et al. (1999).

3) Many of the figure legends are confusing. For instance, in Fig. 4, it is necessary to read the whole legend to find out that the first 6 panels are for SST and the last 6 panels are for SSH.

---

## Referee Comment (RC2) · Anonymous Referee #2 · 20 May 2019

In this article, the authors present a thorough procedure for generating physical-biogeochemical ensembles using various stochastic parameterization techniques. The ensembles are verified using observations such as SST, SLA, chlorophyll (ocean color) and later tested in a data assimilation framework. The presented research can be useful however, my major concern is the relevancy of the paper to GMD journal. There is technically no model development but rather a detailed application of stochastic techniques for generating and studying ocean variability. Overall, the paper is quite long and there are lots of sections that seem disconnected sometimes making it really diffi-

cult to follow. It took me a long time to read through and I'm disappointed for not being able to prepare a better review. It's just hard to grasp on everything. One other issue is structure and the flow of the manuscript, it felt more like a report rather than a directed scientific study. There is probably some novelty in the choice of the stochastic schemes but I wonder if that's enough for justifying this lengthy article. In general, if this to be reconsidered I strongly encourage a detailed polish so the procedure could look more clear, concise and easier to follow.

---

## Author Comment (AC1) · 16 Jun 2019

**Response to reviewer #1**

The response to the reviewer is structured as follows: **RC:** comments from the reviewer, **AC:** author's response, **Changes:** author's prepared changes for the revised manuscript.

> **RC:** The general purpose of this paper is to produce ensemble simulations with a physical-biogeochemical model of the Bay of Biscay, to assess these simulations using sea surface temperature, sea level anomaly and ocean color observations, and to evaluate the potential impact of the new ensembles in a Kalman data assimilation framework. In terms of model development, the novelty is in the introduction of long-range horizontal correlations in the stochastic parameterization of the NEMO ocean model. In terms of application, many ensemble simulations are performed to evaluate the effect of a variety of stochastic perturbations. In my view, the general approach is meaningful, even if the paper is clearly more directed towards practical application than model development.
>
> However, for the reasons explained below, I struggled a lot to read the paper and to write a review, mainly because there are so many different things involved (as acknowledged in the introduction: "the scientific objectives cover a broad spectrum of interdisciplinary components"), and because the text is often unclear and makes it difficult to figure out what is effectively done (despite the many details provided). I also found that the methods used to simulate the large-scale perturbations and to evaluate the results are questionable, without sufficient explanations and justifications. Overall, I do not know if these problems can be solved or not, but this should certainly require a thorough reconsideration of the text to improve the clarity of the arguments (more simple, more factual), and to provide the missing explanations and justifications.

**AC:** We thank the reviewer for the constructive comments and suggestions aimed at improving the manuscript. In the revised manuscript, we will provide all necessary details addressing the reviewer's main comments.

Please note that we propose to reframe the whole manuscript as concentrating on the assessment of a stochastic model composed of (NEMO + updated stochastic module); this is detailed below in our response to main comment 3. We believe that in that way the sections will follow each other more naturally, and that paper will better fit the GMD NEMO Special Issue.

We list here the main development components of this study:

- The NEMO stochastic modules are complemented to introduce long-range spatial correlations in a high-resolution coastal/regional configuration.
- An ensemble-based toolbox is developed within the SDAP data assimilation framework and interfaced with NEMO, in order to:
  - provide empirical consistency diagnostics measuring the ensemble skill with respect to data – in effect, these diagnostics measure the skill of the stochastic model composed of (NEMO + updated stochastic module).
  - generate incremental analyses for data assimilation.
- The NEMO namelist is enriched to perturb several other physical-biogeochemical variables, providing also an updated stopar.F90 module.

**Changes:** We notably mention the most important changes prepared for the revised manuscript, following the reviewer's main comments:

- Section 2.2 will be re-written to better describe the new method calculating large-scale perturbations. We are also considering to include new figure(s) illustrating perturbation patterns based on different methods.
- Section 3.2 will be re-written to give formal expressions for the diagnostics used in the ensemble consistency analysis with respect to observations (stochastic model verification diagnostics). Figures 6 and 8 of the original manuscript will be updated to illustrate regionalization.
- We will more precisely focus the manuscript, improve its flow, polish it whole, and explain the reasons of the various steps we chose to take at the beginning of each section. For this, we intend to better position the revised manuscript with respect to the aims and scope of the NEMO Special Issue.
* * *
**Main comments**

**RC:** 1) I do not understand the method used to generate the large-scale perturbations. I do not know what an "elliptic Gaussian equation" is. I checked the expression in google, and I had only 7 replies, all of them (except one) from the author's work. A classic approach to generate a long-range correlated noise from a white noise is to solve an elliptic partial differential equation like E(x) = w, where w is a white noise, x the resulting correlated noise, E an elliptic operator, like a Laplacian operator (times the square of a length scale). I checked the code provided with the manuscript and I found no solver for an elliptic partial differential equation. Anyway, whatever the method, we need to understand what it is, how it works, what are the benefits with respect to other methods, and what are the expected properties of the resulting noise in terms of correlation structure. If not, there is no real model development in this paper, and it is probably better to move to another journal.
* * *
**AC:** The reviewer's criticism is correct. We apologize for not giving the proper description of the method to generate long-range spatial correlations. In fact, we wish to thank the reviewer for spotting this ambiguity in the text, which undermined the presentation of the model development work presented in this study.

The model development discussed here is aimed at introducing long-range correlations, based on a horizontal structure of covariances derived from the bivariate Gaussian function.

More specifically, we generate equal probability density contours, using a bivariate normal distribution function over a few random grid-points in the model domain. Hereafter, the terms normal and Gaussian can be used interchangeably. The centers of the distributions are called modes. For each distribution, the spatial covariance matrix is positive definite with non-equal length-scale variances and therefore, the function is said to be elliptic[1]. At each random grid-point we compute a 2D unimodal pdf. In order to generate multimodal patterns, we use a linear combination of the unimodal pdfs. The multimodal pdf is rescaled to obtain the desired
* * *
[1] It was our fault not giving formal expressions for the method used in this study to introduce long-range spatial correlations, that perhaps confused the reviewer expecting to see an elliptic partial differential equation similar to the Laplacian operator.

uncertainty amplitude across the ensemble members and subdomains (e.g. 30% for the wind perturbation).

Fortran codes to calculate horizontal structures using the Gaussian function, is also available through the SANGOMA project (cf. sangoma-tools in http://www.data-assimilation.net/). The code provided in zenodo is compatible with the ensemble capabilities of NEMO using the MPI double parallelization domain (https://zenodo.org/record/2556530#.XQSjRi2B01g). Our method yields similar patterns in terms of variable correlation length-scales, with the ones presented by Barth et al., (2009) (cf. their Fig.2).

In this study, we have used the Laplacian operator exactly as it is suggested by the reviewer. This capability is already implemented in NEMO, cf. subroutine sto_par_flt(psto) and function sto_par_flt_fac(kpasses) inside stopar.F90 module. The Gaussian function complements the Laplacian operator, including in the namelist the option to choose between the two approaches. In section 2.2 of the original manuscript, we have discussed that the Laplacian operator is mainly effective in coarse resolution configurations implementing a few passes. We have actually demonstrated that the option to iterate too many times the operator in BISCAY36 (e.g. 100 passes) is not optimal and shows noisy patterns (Fig. 2h). At this point, we wish to apologize for an erratum spotted in the text of the original manuscript (page 5, line 28), writing by mistake Fig. 2g instead of the correct subplot Fig. 2h, which perhaps confused the reviewer regarding the implementation of the Laplacian operator in this study. Finally, we have found that there is a technical limitation on the number of Laplacian passes one can perform, with an increased risk of a model crash.

**Changes:** In the revised manuscript, we will include in section 2.2 a detailed description of the method used to introduce long-range correlations. The phrase "elliptic Gaussian equation" will be removed and replaced where appropriate by similar expression(s) to horizontal structures based on the "bivariate Gaussian function". In Appendix A we intend to discuss the challenges implementing the Gaussian function in the NEMO MPI double parallelization domain, since long-range spatial correlations per member can span several subdomains in the model domain (the Laplacian operator smooths neighboring grid-points per pass). Finally, we are considering to include new figure(s) in the text and/or as supplementary material, illustrating perturbation patterns and/or ensemble spreads based on different methods.

**RC:** 2) I do not really understand the method that is used to compare the stochastic ensemble simulations to observations. This is at the core of the paper and would need to be based on a solid ground. The results of the comparisons are displayed in Figs. 6, 7 and 8, which mainly show averages or mixtures over the whole domain. On the one hand, what is done to compute these statistics is unclear to me, and it is difficult to have a clear idea of what stands behind the various curves that are shown. On the other hand, I do not understand how these global results can be used to deduce that the ensemble are more or less consistent with the observations. It is indeed repeatedly stated in the paper that the system is very heterogeneous, and I thus presume that the local consistency with observations is an important issue. In summary, we need to understand the logic supporting the validation procedure, and why it is applicable to this problem.

**AC:** We take note for the helpful feedback. We are going to give the formal expressions for the diagnostics used in the ensemble consistency analysis in the revised

manuscript. Section 3.2 will be re-written to elaborate what is done to compute model-data misfits in an ensemble-based framework. For reference, the corresponding code is available here:

https://sourceforge.net/p/sequoia-dap/code/HEAD/tree/branches/1.6/components/SDAP-beluga-ulib-NEMO_3.6_scrumcat-ver1.6/u_scrum_basic.F90

The reviewer also rightfully raises the issue of local dependency of ensemble consistency results, since the system is indeed spatially heterogeneous. In order to properly address this comment, we will update Figures 6 and 8 of the original manuscript to illustrate regionalization of metrics. We aim at discussing two distinct areas in the Bay of Biscay, namely the Armorican shelf and the Abyssal plain, both governed by different physical-biogeochemical processes.

Finally, let us remark that (1) the ensemble consistency verification module developed for this study is another new development for NEMO, and that (2) ensemble-based diagnostics are used to measure the skill of the stochastic model with respect to data, which is within the scope of the NEMO Special Issue.

**Changes:**

- Section 3 will be more precisely focused, in particular in regard for stochastic model (ensemble) skill evaluation.
- Formal expressions of ensemble-based diagnostics will be given in an attempt to better explaining what is done to evaluate the stochastic model skill.
- Figures 6 and 8 will be updated to illustrate the regionalization of metrics, followed by a discussion.

**RC:** 3) The paper is made of three different components (description and development of the method, evaluation of the ensemble simulations, and impact on data assimilation), which are presented and discussed almost independently. I understand that correlated stochastic perturbations were needed to produce the ensemble, and that good ensembles are likely to improve data assimilation. I think this is not sufficient to connect the components of the paper together. For instance, the effect of the new methodological development on the ensemble simulations is not specifically discussed, and the effect of data assimilation is mainly described in terms of realism and physical consistency than in relation to what is done in the rest of the paper.

**AC:** We agree with the reviewer and we think we can present our results in a tighter and more focused manner, while better fitting within the aims and scope of the NEMO Special Issue. The structure of the three main components mentioned by the reviewer (i.e. sections 2, 3 and 4) will be placed in the general framework of stochastic model evaluation, in a high-resolution coastal/regional configuration for the Bay of Biscay. We will also better explain the reasons of each step at the beginning of each section.

Not only NEMO in this high-resolution coastal/regional configuration is evaluated, so we updated the stochastic module, and we evaluate in this paper the skill of the stochastic model composed of (NEMO + updated stochastic module):

- The updated stochastic module (Section 2), incorporating changes for long-range correlation scales, and the ensemble-based consistency analysis module (Section 3), are new component developments for NEMO.
- The stochastic model skill is quantitatively evaluated with respect to physical-biogeochemical data (Section 3).
- The stochastic model is qualitatively evaluated by using its output to generate incremental analyses, in an ensemble-based data assimilation framework (Section 4).

We think that this (1) is faithful to our original intentions, (2) is a suitable angle to connect the main sections of the manuscript, and (3) better fits the aims and scope of the NEMO Special Issue.

**Changes:** In the revised manuscript, we will reframe Sections 2, 3 and 4 using the above. Several parts of the text will be re-written, especially the introductory paragraphs of each section and perhaps also some of their titles, to better connect the technical and scientific components of this study.

**Other comments**

**RC:** 1) A previous paper by Vervatis et al. (2016) is cited throughout the paper. I think it would be necessary to better position the present paper with respect to the previous one.

**AC:** We thank the reviewer for the useful suggestion. In the revised manuscript, we will better position the present study with respect to our previous work (Vervatis et al., 2016).

**RC:** 2) The appropriate reference for the SPPT scheme is Buizza et al. (1999).

**AC:** The reference will be mentioned in the appropriate place in the revised manuscript.

**RC:** 3) Many of the figure legends are confusing. For instance, in Fig. 4, it is necessary to read the whole legend to find out that the first 6 panels are for SST and the last 6 panels are for SSH.

**AC:** We have changed some of the figure legends in the revised manuscript and reads better now.

**References mentioned in this response letter**

Barth, A., Alvera-Azcárate, A., Beckers, J.-M., Weisberg, R. H., Vandenbulcke, L., Lenartz, F., and Rixen, M.: Dynamically constrained ensemble perturbations-application to tides on the West Florida Shelf, Ocean Sci., 5, 259-270, https://doi.org/10.5194/os-5-259-2009, 2009.

Buizza, R., Miller, M., and Palmer, T.N.: Stochastic representation of model uncertainties in the ECMWF ensemble prediction system. Quart. J. Roy. Meteor. Soc., 125, 2887-2908, doi:10.1002/qj.49712556006, 1999.

Vervatis, V., C.E. Testut, P. De Mey, N. Ayoub, J. Chanut, and Quattrocchi, G.: Data assimilative twin-experiment in a high-resolution Bay of Biscay configuration: 4D EnOI based on stochastic modelling of the wind forcing. Ocean Modelling, 100, 1-19, http://dx.doi.org/10.1016/j.ocemod.2016.01.003, 2016.

---

## Author Comment (AC2) · 16 Jun 2019

**Response to reviewer #2**

The response to the reviewer is structured as follows: **RC:** comment from the reviewer, **AC:** author's response, **Changes:** author's prepared changes for the revised manuscript.

> **RC:** In this article, the authors present a thorough procedure for generating physical-biogeochemical ensembles using various stochastic parameterization techniques. The ensembles are verified using observations such as SST, SLA, chlorophyll (ocean color) and later tested in a data assimilation framework. The presented research can be useful however, my major concern is the relevancy of the paper to GMD journal. There is technically no model development but rather a detailed application of stochastic techniques for generating and studying ocean variability. Overall, the paper is quite long and there are lots of sections that seem disconnected sometimes making it really difficult to follow. It took me a long time to read through and I'm disappointed for not being able to prepare a better review. It's just hard to grasp on everything. One other issue is structure and the flow of the manuscript, it felt more like a report rather than a directed scientific study. There is probably some novelty in the choice of the stochastic schemes but I wonder if that's enough for justifying this lengthy article. In general, if this to be reconsidered I strongly encourage a detailed polish so the procedure could look more clear, concise and easier to follow.

**AC:** We thank the reviewer for the general comments to revise the manuscript. We take note that we should more precisely focus the manuscript, improve its flow, polish it whole, and explain the reasons of the various steps we chose to take at the beginning of each section.

Most of the comments between the two reviewers are common and can be addressed by similar actions in revised manuscript. Below, we iterate parts of our response given also to the other reviewer.

Please note that we propose to reframe the whole manuscript as concentrating on the assessment of a stochastic model composed of (NEMO + updated stochastic module). We believe that in that way the sections will follow each other more naturally, and that paper will better fit the GMD NEMO Special Issue.

We agree with the reviewer that we can present our results in a tighter and more focused manner. The structure of the three main components (i.e. sections 2, 3 and 4) will be placed in the general framework of stochastic model evaluation, in a high-resolution coastal/regional configuration for the Bay of Biscay.

Not only NEMO in this high-resolution coastal/regional configuration is evaluated, so we updated the stochastic module, and we evaluate in this paper the skill of the stochastic model composed of (NEMO + updated stochastic module):

- The updated stochastic module (Section 2), incorporating changes for long-range correlation scales, and the ensemble-based consistency analysis module (Section 3), are new component developments for NEMO.
- The stochastic model skill is quantitatively evaluated with respect to physical-biogeochemical data (Section 3).
- The stochastic model is qualitatively evaluated by using its output to generate incremental analyses, in an ensemble-based data assimilation framework (Section 4).

We think that this (1) is faithful to our original intentions, (2) is a suitable angle to connect the main sections of the manuscript, and (3) better fits the aims and scope of the NEMO Special Issue.

**Changes:** In the revised manuscript, we will reframe Sections 2, 3 and 4 using the above. Several parts of the text will be re-written, especially the introductory paragraphs of each section and perhaps also some of their titles, to better connect the technical and scientific components of this study.